# Core–Shell Catalysts for Conventional Oxidation of Alcohols: A Brief Review

Luís M. M. Correia [1,2], Maxim L. Kuznetsov [1] and Elisabete C. B. A. Alegria [1,2,*]

[1] Centro de Química Estrutural, Departamento de Engenharia Química, Instituto Superior Técnico, Universidade de Lisboa, 1049-001 Lisboa, Portugal

[2] Departamento de Engenharia Química, Instituto Superior de Engenharia de Lisboa, Instituto Politécnico de Lisboa, 1959-007 Lisboa, Portugal

* Correspondence: elisabete.alegria@isel.pt; Tel.: +351-218317000

**Abstract:** This review highlights recent research on the application of core–shell structured materials as catalysts in the oxidation of alcohols to value-added products, such as benzaldehyde, acetophenone, benzophenone, cinnamaldehyde, and vanillin, among others. While the application of various unconventional energy inputs (such as microwave and ultrasound irradiation) was reported, this paper focuses on conventional heating. The oxidation of homocyclic aromatic, heterocyclic aromatic, aliphatic, and alicyclic alcohols catalyzed by core–shell composite catalysts is addressed. This work also highlights some unique advantages of core–shell nanomaterial catalysis, namely the flexibility of combining individual functions for specific purposes as well as the effect of various parameters on the catalytic performance of these materials.

**Keywords:** core–shell structure; nanocomposites; oxidation of alcohols; catalysis; conventional synthesis



## 1. Introduction

Composites are hybrid materials that combine two or more elements, usually with different physical or chemical properties, with enhanced properties relative to the individual properties of the components. These materials include nanocomposites, where one of the constituents has a nanoscale morphology ($\leq$100 nm) and exhibits a core–shell structure (inner material as a core and outer layer material as a shell) with a wide range of inorganic@inorganic, inorganic@organic, organic@inorganic, and organic@organic combinations [1]. Inorganic components comprise metals, metal sulfides, metal oxides, etc., while organic ones include three-dimensional network polymers or other carbon-containing species such as carbon nanotubes, graphene, etc. Core–shell structures with both core and shell are made of inorganic materials and are the most explored category, with a wide range of applications, namely in catalysis, optoelectronics, and bio-sensing [1].

The core–shell structures can exhibit different morphologies, depending on their shape and composition. The first classification takes into account the number of cores and shells in nanocomposite structures (Figure 1) and includes four types, i.e., (i) conventional single core–shell structures bearing one core and one shell, (ii) multicore–shell structures (several cores, one shell), (iii) core–multishell structures also known as nanomatryoshka (one core, several shells), and (iv) multicore–multishell structures (several cores and shells) [2].

The core–shell morphology can be divided into two categories. In the first one, both the core and the shell have the same shape, whereas in the second one, they have different shapes (Figure 2). In the first case, which is more common, the shell is usually deposited on top of the already synthesized core, thus retaining its original shape. If the thickness and uniformity of the shell are not controlled, a composite with two different parts (core–shell) can be obtained. This type of structure can be formed due to self-assembly and crosslinking of polymeric materials [3].

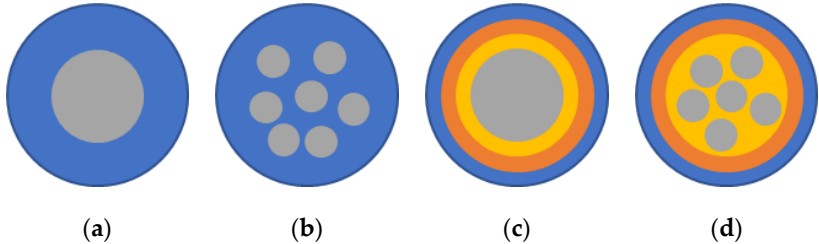

**Figure 1.** Types of core–shell structures based on morphology: conventional single core–shell (**a**), multicore–shell (**b**), core–multishell (**c**), and multicore–multishell (**d**) structures.

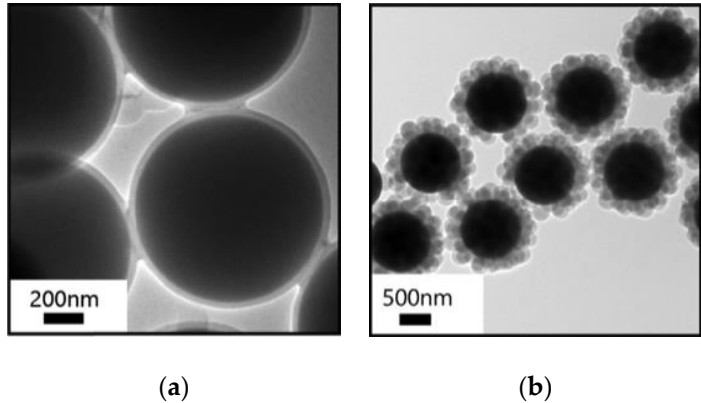

**Figure 2.** TEM images of $SiO_2$-MAPTMS@PMMA core–shell particles with both core and shell spherical shapes (**a**) and of $SiO_2$-MAPTMS@PS core–shell particles with different core and shell shapes, spherical and flower, respectively (**b**). Reproduced from [4] with permission from the Walter de Gruyter GmbH.

Hybrid functional materials such as metal-organic frameworks (MOFs) have been explored in core–shell structures, acting either as a core or as a shell and leading to numerous possible combinations, e.g., silica@polymer [4], metal@metal oxide [5], polymer@polymer [6], silica@MOF [7], and MOF@COF [8], among others.

Core–shell nanocomposites are known for their high functionality, as they can improve or even acquire new physical and chemical properties, making them suitable for various applications. However, these materials have a limited storage capacity, and to overcome this weakness, hollow composites without any core and with an empty shell have emerged (Figure 3). Finally, the so-called yolk–shell carbon nanostructures represent an intermediate type between the hollow and core–shell materials. They show improved catalytic properties, have all the advantages of the core structures, and have a significant storage capacity due to the free space between the core and the shell [9,10].

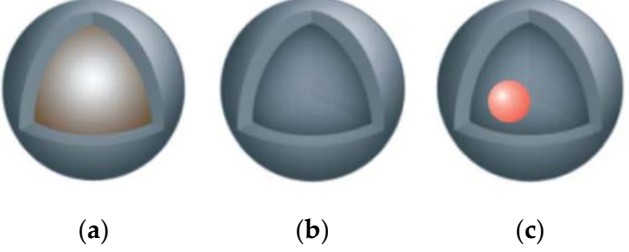

**Figure 3.** Composites of the core–shell (**a**), hollow core–shell (**b**) and yolk–shell (**c**) types. Adapted from [10] with permission from the Royal Society of Chemistry.

The structured nanocomposites can be synthesized using different approaches such as sol-gel [11], precipitation [12], electrochemical deposition [13], successive ionic layer

adsorption and reaction (SILAR) [14], layer by layer [15], polymerization [16], template-assisted approach [17], self-assembly [18], and reduction [19], among others.

Core–shell composites are materials with tremendous potential in several areas of application such as biomedicine (drug delivery [20], biosensor [21], and bioimaging [22]), energy storage and capacity (hydrogen storage [23], supercapacitors [24], and solar cells [25]), photonics [26], catalysis (photocatalysis [27], electrocatalysis [28]), etc.

The selective oxidation of alcohols plays an important role in various industrial processes related to food, perfume, and pharmaceutical production, as well as in chemical synthesis. This review article focuses on the application of the core–shell composites in catalysis, particularly in the conventional oxidation of alcohols to value-added products such as benzaldehyde, acetophenone, benzophenone, cinnamaldehyde, and vanillin, among others. The next four sections discuss the oxidation of aromatic homocyclic, aromatic heterocyclic, aliphatic, and alicyclic alcohols catalyzed by core–shell composites. Recent studies have mainly focused on the selective oxidation of aromatic alcohols and their derivatives and much less on aliphatic alcohols which could be explained by the higher reactivity of CH bonds in the alcohols bearing aromatic substituents [29–31].

## 2. Oxidation of Aromatic Homocyclic Alcohols

### 2.1. Benzylic Alcohols

In 2015, Wang et al. synthesized the Au@Pd bimetallic core–shell supported on $SiO_2$ and investigated it as a catalyst for the solvent-free selective aerobic oxidation of benzyl alcohol (Scheme 1) [32]. The thickness of the Pd layer controlled by atomic layer deposition (ALD) at a rate of ~0.8 nm per Pd ALD cycle affects the catalytic activity. The total yield increases and the benzaldehyde selectivity decreases with the number of cumulative Pd ALD cycles reaching the maximum turnover frequency (TOF) of $2.76 \times 10^4$ $h^{-1}$, at the optimized Pd shell thickness of 0.6–0.8 nm obtained after 8 Pd ALD cycles (Au@8Pd/$SiO_2$) (entry 1, Table 1) [32].

**Scheme 1.** Oxidation of benzyl alcohol to benzaldehyde and benzoic acid catalyzed by Au@Pd bimetallic core–shell structures supported on $SiO_2$ [32].

In 2020, Luo et al. reported the Au/SH-MON@m-$SiO_2$ core–shell which consists of mesoporous organosilica functionalized with thiol groups (shell) coating gold nanoparticles housed as an internal part of the catalyst (core). This catalyst displays cooperative properties, i.e., while the mesopores of the shell facilitate access and release of reactants and products, and the internal pores of the organosilica provide a hydrophobic microenvironment [33]. The catalytic activity of this core–shell was tested for the aerobic oxidation of benzyl alcohol and, although the reported conversion was not impressive (possibly due to the low gold content of 0.91 wt% Au), the selectivity towards benzaldehyde is remarkable (99%, entry 2, Table 1). These core–shell structures showed rather low stability during the recycling process. A significant loss of activity is already observed at the end of the third cycle (ca. 41% compared to the first cycle) while maintaining a high selectivity towards benzaldehyde (99%). These results have been attributed to the degradation of the mesoporous silica shell and the aggregation of the gold nanoparticles, which affect the active sites and, thus, compromise their catalytic activity [33].

At the same time, Hammond-Pereira et al. aerobically oxidized benzyl alcohol in the presence of silica-encapsulated gold core@shell nanoparticles, Au@m-$SiO_2$, under alkaline conditions using $K_2CO_3$. The substrate conversion of 60.4% was obtained after 1 h of reaction under solvent-free conditions, and a TOF value of $1.28 \times 10^4$ $h^{-1}$ was registered, indicating a rather high activity [34]. The selectivity for aldehyde (ca. 75%) (entry 3, Table 1) is attributed to the mesoporous shell of the AuNPs which seems to inhibit the formation of

larger species. The addition of potassium carbonate has a distinct effect on the conversion and selectivity of the reaction, i.e., it favors the substrate conversion from 17.3 to 60.4% but decreases the selectivity from 98.7 to 75.0% [34].

**Table 1.** Catalytic results for benzyl alcohol oxidation with different core–shell nanostructures as catalysts.

| Entry | Catalyst | Yield (%) | Selectivity (%) | Ref. |
|---|---|---|---|---|
| 1 [a] | Au@8Pd/SiO$_2$ | 91 | 87 | [32] |
| 2 [b] | Au/SH-MON@m-SiO$_2$ | 51 | 99 | [33] |
| 3 [c] | Au@m-SiO$_2$ | 60 | 75 | [34] |
| 4 [d] | Pd/ZDC@m-SiO$_2$ | 99 | 99 | [35] |
| 5 [e] | Fe$_2$O$_3$@Fe$_2$O$_3$ | 42 | 95 | [36] |
| 6 [f] | $\gamma$-Fe$_2$O$_3$@Ni$_3$Al-LDH@Au$_{25}$-0.053 | 99 | 99 | [37] |
| 7 [g] | Fe$_3$O$_4$@PAH@PSS@Pd | 95 | 90 | [38] |
| 8 [h] | Fe$_3$O$_4$@SiO$_2$@Au@Pd | 88 | 86 | [39] |
| 9 [i] | SiO$_2$@Co$_3$O$_4$@Au@m-SiO$_2$ | 58 | 82 | [40] |
| 10 [j] | Co$_3$O$_4$@Au@m-SiO$_2$ | 55 | 84 | [41] |
| 11 [k] | PILM@Pd@CeO$_2$ | 48 | 98 | [42] |
| 12 [l] | PAM@SiO$_2$@Au | 40 | 90 [n] | [43] |
| 13 [m] | PS@PNIPA/Au | - | >99 | [44] |

[a] Reaction conditions: benzyl alcohol (48 mmol), 20 mg of Au@8Pd/SiO$_2$, 90 °C, 6 h under O$_2$ atmosphere (15 mL/min); [b] benzyl alcohol (0.48 mmol), 60 mg of Au/SH-MON@m-SiO$_2$, toluene (5 mL), K$_2$CO$_3$ (0.48 mmol), 90 °C, 12 h under high air flow rate; [c] benzyl alcohol (9.6 mmol), Au@m-SiO$_2$ (0.5 µmol Au), K$_2$CO$_3$ (0.94 mmol), 100 °C, 1 h under 6 bar O$_2$; [d] benzyl alcohol (0.5 mmol), 30 mg of Pd/ZDC@m-SiO$_2$, toluene (1 mL), 100 °C, 12 h under atmospheric air; [e] benzyl alcohol (10 mmol), Fe$_2$O$_3$@Fe$_2$O$_3$ (1 mol% of iron content), 75 °C, 12 h, and H$_2$O$_2$ 30% aq. sol. (10 mmol) as oxidant; [f] benzyl alcohol (1 mmol), $\gamma$-Fe$_2$O$_3$@Ni$_3$Al-LDH@Au$_{25}$-0.053 (Au: 0.115%), toluene (5 mL), 80 °C, 1.5 h under 1 atm O$_2$ (20 mL/min); [g] benzyl alcohol (20 mmol), 100 mg of Fe$_3$O$_4$@PAH@PSS@Pd, water (40 mL), 80 °C, 6 h and H$_2$O$_2$ 30% aq. sol. (3 mL) as oxidant; [h] benzyl alcohol (9.6 mmol), Fe$_3$O$_4$@SiO$_2$@Au@Pd (3.4 µmol Au, 0.34 µmol Pd), 100 °C, 2.5 h under 6 bar O$_2$; [i] benzyl alcohol (76.9 mmol), 40 mg of SiO$_2$@Co$_3$O$_4$@Au@m-SiO$_2$, K$_2$CO$_3$ (4.3 mmol), 160 °C, 6 h under O$_2$ (60 mL/min); [j] benzyl alcohol (76.9 mmol), 40 mg of Co$_3$O$_4$@Au@m-SiO$_2$, K$_2$CO$_3$ (4.3 mmol), 140 °C, 5 h under O$_2$ (50 mL/min); [k] benzyl alcohol (20 mmol), 25 mg of PILM@Pd@CeO$_2$, water (40 mL), K$_2$CO$_3$ (4 mmol), 160 °C, 6 h under O$_2$ (60 mL/min); [l] benzyl alcohol (0.2 M), 250 mg of PAM@SiO$_2$@Au, methanol/water (50/50 V/V%), NaOH (0.3 M), 60 °C, 20 h; [m] benzyl alcohol (9 mmol), 1 mL of PS@PNIPA/Au (0.1 wt% of metal-microgel composite), water (8 mL), K$_2$CO$_3$ (3 mmol), 40 °C, 20 h under 1 atm O$_2$; [n] selectivity to benzoic acid.

Zhang et al. reported the successful preparation of a Pd/ZDC@m-SiO$_2$ core–shell structure, which consists of Pd nanoparticles (PdNPs) supported on ZIF-8 derived porous carbon (ZDC) as a core and mesoporous silica as a shell, and its application as a catalyst in the aerobic oxidation of benzyl alcohol. Excellent catalytic performance with high conversion and selectivity was observed (entry 4, Table 1). The high stability of this core–shell was confirmed by reusing it for eight cycles without loss of activity and was attributed to its structural configuration which prevents the sintering of PdNPs during the calcination process as well as their leaching during catalytic tests [35].

Lian et al. synthesized the homogeneous spherical hematite core–shell (Fe$_2$O$_3$@Fe$_2$O$_3$) through a one-pot surfactant-free solvothermal treatment of FeCl$_3$.6H$_2$O and investigated its catalytic activity in the peroxidative oxidation (H$_2$O$_2$ 30% aq. sol.) of benzyl alcohol achieving 42% conversion and 95% selectivity towards benzaldehyde after 12 h of reaction (entry 5, Table 1) [36]. Other catalysts such as microparticles of $\alpha$-Fe$_2$O$_3$, $\gamma$-Fe$_2$O$_3$, and Fe$_3$O$_4$ (1 mol% Fe) have been tested under the same reaction conditions. The selectivities towards benzaldehyde remained of the same order for the core–shell (92–94%), but the benzyl alcohol conversion was found to be ca. 5 times lower (7–9%), demonstrating the superiority of the core–shell structure. The core–shell hematite catalyst Fe$_2$O$_3$@Fe$_2$O$_3$ proved to be very promising since it allowed the combination of high catalytic activity with high stability since the catalytic performance remained the same after 5 catalytic cycles (41% yield and 95% selectivity for benzaldehyde) [36].

Yin et al. screened the catalytic activity of the hierarchical hollow nanostructured core–shell $\gamma$-Fe$_2$O$_3$@Ni$_3$Al-LDH@Au$_{25}$-0.053 in the aerobic oxidation of benzyl alcohol [37] and showed an excellent performance with almost complete conversion observed after

1.5 h (entry 6, Table 1). Comparing only the percentage yields presented in Table 1, this catalyst and the Pd/ZDC@m-SiO$_2$ synthesized by Zhang et al. [35] showed the best catalytic conversion of benzyl alcohol under specific reaction conditions [37].

Zhang et al. reported the synthesis of core–shell microspheres using Fe$_3$O$_4$ as a core coated with layers of poly(allylamine hydrochloride)-Pd(II) (PAH) and poly(sodium 4-styrenesulfonate) PSS and by an outer layer of palladium nanoparticles as the shell, denoted here as Fe$_3$O$_4$@PAH@PSS@Pd [38]. This catalyst was used in the peroxidative oxidation reaction (H$_2$O$_2$ 30% aq. sol.) of benzyl alcohol with an alcohol conversion of 95% and selectivity towards benzaldehyde of 90% (entry 7, Table 1). The microspheres could be reused for three more cycles without significant loss of activity [38].

The synthesis of Fe$_3$O$_4$@SiO$_2$@Au@Pd core–shell and its catalytic activity in the aerobic oxidation reaction of benzyl alcohol was reported by Silva et al. [39]. The optimized Au:Pd molar ratio of 10:1 resulted in the 87.7% conversion of benzyl alcohol with the 86.4% selectivity towards benzaldehyde (entry 8, Table 1). The stability of Fe$_3$O$_4$@SiO$_2$@Au@Pd was also evaluated, but the results showed fluctuations in the conversion between cycles. This effect was attributed to the presence of amino groups in Fe$_3$O$_4$@SiO$_2$ used for a better impregnation of gold in the core–shell which could interact with the reaction products. Meanwhile, at the end of the sixth cycle, the conversion was 50.3%, and the selectivity towards benzaldehyde increased to 92.5% [39].

More recently, Silva et al. calcinated the Fe$_3$O$_4$@SiO$_2$@Au@Pd core–shell and observed that the thermal treatment altered the structure of the nanocomposite, leading to an increase in particle size, metal segregation, and oxidation of palladium to PdO. The treated material showed increased catalytic activity (4-fold) for the aerobic oxidation of benzyl alcohol at 100 °C under 6 bar O$_2$ with a maximum activity for an Au:Pd molar ratio of 1:2 [45].

In 2019, Lv et al. reported the SiO$_2$@Co$_3$O$_4$@Au@m-SiO$_2$ core–shell [40] while Yuan et al. synthesized the hollow Co$_3$O$_4$@Au@m-SiO$_2$ core–shell [41], both being applied in the aerobic oxidation reaction of benzyl alcohol. After optimizing the reaction parameters such as the amount of catalyst, temperature, oxygen flow rate, and reaction time, a conversion of 58% and a selectivity towards benzaldehyde of 82% were obtained for the former catalyst after 6 h reaction at 160 °C and a 60 mL/min O$_2$ flow rate, whereas for the latter core–shell, a conversion and a selectivity of 55 and 84%, respectively, were achieved after 5 h reaction at 140 °C and a 50 mL/min O$_2$ flow rate (entries 9 and 10, Table 1). Both catalysts demonstrated a good resistance maintaining the catalytic performance after 5 cycles [40,41].

In 2020, Wu et al. synthesized polymeric ionic liquid microspheres (PILM) (core) and coated them with a layer of Pd nanoparticles (PdNPs) covered with a layer of CeO$_2$ (shell) forming the PILM@Pd@CeO$_2$ core–shell [42]. The supported PdNPs were uniformly distributed on the surface of the polymer microspheres with aggregation being avoided through the addition of a CeO$_2$ layer. The core–shell was used for the aerobic oxidation reaction of benzyl alcohol; under optimized reaction conditions in terms of reaction parameters such as the amount of an alkaline additive (K$_2$CO$_3$), temperature (160 °C), and oxygen flow rate (60 mL/min), an alcohol conversion of 48%, a selectivity towards benzaldehyde of 98%, and a TOF of $6.33 \times 10^2$ h$^{-1}$ were achieved (entry 11, Table 1). They also tested the stability of the catalyst using recycling tests and found that these core–shell structures maintained their catalytic performance after five cycles [42].

In 2023, Su et al. synthesized the spherical core–shell PILM@Au@Al(OH)$_3$ with the polymeric ionic liquid microspheres (PILM) as the core coated with gold nanoparticles and covered with an aluminum hydroxide layer. This material was used as a catalyst in the peroxidative oxidation reaction (H$_2$O$_2$, 30% aq. sol.) of benzyl alcohol. The optimized reaction conditions are 30 mg of catalyst, 2.2 wt% of Au content, 80 °C, H$_2$O$_2$:benzyl alcohol molar ratio of 1.25:1, and 6 h of the reaction time. Under these conditions, a conversion of 76% and a selectivity towards benzaldehyde of 92% were obtained. The effect of recycling the catalyst was also studied. It was found that after five cycles, the core–shell was stable and did not show significant loss in conversion and selectivity [46].

Meaney et al. synthesized the PAM@SiO$_2$@Au core–shell which consisted of poly(acrylamide) with deposited SiO$_2$ layer (shell) and gold nanoparticles (AuNPs) on the surface. It was then applied in the oxidation of benzyl alcohol and showed a maximum conversion of ca. 40% and a selectivity towards benzoic acid of 90% after 20 h of reaction (entry 12, Table 1). The effect of the reaction time on the product yields was analyzed, and benzoic acid was already detected as the main product after 8 h. The presence of a strong base (NaOH) promoted the formation of benzoic acid as the main product [43].

Lu et al. synthesized the gold–microgel nanocomposite PS@PNIPA/Au core–shell consisting of a spherical polystyrene core coated with a poly(*N*-isopropylacrylamide) (PNIPA) shell cross-linked by *N,N*′-methylenebisacrylamid [44]. This material was subsequently used as a support for the immobilization of gold nanoparticles and applied as a catalyst for the aerobic oxidation reaction of benzyl alcohol under mild conditions (entry 13, Table 1). Green chemistry principles were considered using water as the solvent and air as the oxidant at room temperature (25 °C). The thermosensitive character of the core–shell microgels is responsible for the temperature effect on the catalytic activity of the gold–microgel composite. In fact, the TOF value of $1.59 \times 10^2$ h$^{-1}$ was achieved at 25 °C, whereas increasing the temperature to 40 °C resulted in the higher TOF value of $2.20 \times 10^3$ h$^{-1}$. Thus, the catalytic activity varies with the polarity and volume of the gold–microgel composites, since at lower temperatures, the thermosensitive microgel network is hydrophilic and swells in water, while at higher temperatures it becomes hydrophobic and shrinks, becoming oil soluble and favoring the transport of hydrophobic benzyl alcohol to the Au nanoparticle surface [44].

Regarding the yields obtained by the different core–shell in the benzyl alcohol oxidation reaction shown in Table 1, it appears that the lowest yields (40–51%) are obtained in the presence of monometallic core–shells (PAM@SiO$_2$@Au, Fe$_2$O$_3$@Fe$_2$O$_3$, and Au/SH-MON@m-SiO$_2$—entries 12, 5, and 2, respectively, Table 1) whereas the polymetallic core–shells (γ-Fe$_2$O$_3$@Ni$_3$Al-LDH@Au25-0.053, Pd/ZDC@m-SiO$_2$, Fe$_3$O$_4$@PAH@PSS@Pd, and Au@8Pd/SiO$_2$—entries 6, 4, 7, and 1, respectively, Table 1) are more effective and show the highest yields (91–99%), probably due to their synergistic properties.

Several publications reported the application of different core–shell type catalysts in the oxidation of substituted benzyl alcohols as well as the effect of the different group's nature and position.

Zhao et al. tested the catalytic activity of Ru@SQ (Ru nanoparticles coated with disodium anthraquinone-2,6-disulfonate) in the aerobic oxidation of benzyl alcohol and its *p*-methyl and *p*-chlorinated derivatives and found that the conversion under the same conditions is independent of the substrate (Table 2) [47]. The high performance and selectivity towards the aldehyde demonstrated by the Ru@SQ core–shell may be related to the fact that the ruthenium nanoparticles (RuNPs) are surrounded by a uniform organic layer, which could prevent the over-oxidation of the respective benzaldehydes to acids [47].

**Table 2.** Catalytic results (yield and selectivity for aldehyde) for the oxidation of various benzyl alcohols using Ru@SQ core–shell as catalyst [47] [a].

| Entry | Substrate | Main Product | Yield (%) | Selectivity (%) |
|:---:|:---:|:---:|:---:|:---:|
| 1 |  |  | 92 | 95 |
| 2 |  |  | 95 | 95 |
| 3 |  |  | 95 | 95 |

[a] Reaction conditions: substrate (1 g), Ru@SQ (50 mg), 100 °C, 6 h under O$_2$.

Sholjaei et al. optimized the reaction conditions for the oxidation of benzyl alcohol in the presence of the core–shell type catalyst RuO$_2$@ZrO$_2$ [48]. Different temperatures

(25 and 80 °C), amount of catalyst (10, 20, and 30 mg), type of oxidant (TBHP, $H_2O_2$, and $O_2$), and type of solvent (acetonitrile, toluene, and dichloromethane) were tested. Quite good conversions and selectivities towards benzaldehyde were achieved (80 and 87.5%, respectively, entry 1, Table 3). It should be noted that the authors managed to carry out three cycles without significant loss of activity, maintaining the conversion of benzyl alcohol at the level of 80%. Subsequently, the same catalyst was applied to different substituted benzyl alcohols (Table 3), the *p*-chlorobenzyl alcohol being found to be the most active substrate in terms of product yield, selectivity, and reaction time (entry 2, Table 3) [48].

**Table 3.** Catalytic results (yield and selectivity for aldehyde) for the oxidation of various benzyl alcohols using $RuO_2@ZrO_2$ as catalyst [48] [a].

| Entry | Substrate | Product | Yield (%) | Selectivity (%) |
|---|---|---|---|---|
| 1 | | | 80 | 88 |
| 2 [b] | | | 95 | 95 |
| 3 | | | 92 | 92 |
| 4 [c] | | | 45 | 76 |

[a] Reaction conditions: substrate (1 mmol), $RuO_2@ZrO_2$ (4 wt%) (20 mg), acetonitrile (5 mL), 80 °C, 0.25 h and TBHP 70% aq. sol. (0.3 mL) as oxidant; [b] 0.08 h; [c] 0.83 h.

Rostami et al. reported the application of the magnetic $Fe_3O_4@SiO_2$ core–shell supported on an oxo-vanadium ephedrine complex in the peroxidative oxidation (TBHP 70% aq. sol.) of benzyl alcohol [49]. Different reaction conditions (i.e., dichloromethane, acetonitrile, water, or polyethylene glycol (PEG) as a solvent, room temperature or 80 °C, amount of the catalyst of 30, 40, or 50 mg) were investigated. After optimization of the reaction conditions, an excellent yield of 98% integrated with a high selectivity for benzaldehyde (>99) was reached after 3 h (entry 1, Table 4). The catalyst was easily recovered and reused until the sixth cycle without significant loss of activity. This catalyst was also tested for the oxidation of other benzylic alcohols, i.e., those with electron donating (entries 4 and 5, Table 4) and electron withdrawing (entries 2 and 3, Table 4) groups. The latter type of alcohol was found to have lower activity. Interestingly, the yield of the corresponding aldehyde upon oxidation of the more sterically hindered benzylic alcohol substituted with the methyl group at the *ortho* position reached 97% (entry 5, Table 4). However, a much longer reaction time (47 h) was required in this case [49].

Ghalavand et al. reported the design of magnetic nanoparticles copper(I) ion complexes with 1,2,3-triazole bridged immobilized on poly(acrylic acid) denoted as the $Fe_3O_4@MAPTMS@PAA@Triazole@Cu(I)$ composite and its catalytic activity for the peroxidative oxidation ($H_2O_2$ 30% aq. sol.) of benzyl alcohol in acetonitrile [50]. The parameters of temperature and Cu(I) content were optimized, and a yield of 85% of benzaldehyde was obtained after 5 h of reaction (entry 1, Table 5). Due to the magnetic properties of $Fe_3O_4$, the catalyst can be easily separated. Its stability has been evaluated in recovery experiments which show that this composite material retains its catalytic activity during up to three cycles. In addition, a hot filtration test showed a low leaching of iron into the solution confirming the heterogeneous nature of the catalyst. The catalytic activity of this composite was extended to substituted benzylic alcohols carrying the electron donor groups (entries 3 and 4, Table 5) or the electron acceptor groups (entries 2, 5, and 6, Table 5). It was found that alcohols of the latter group are oxidized in a shorter period of time (2 h, 60 °C) compared to those containing the donor groups (5 h, 60 °C or 3 h, 85 °C, entries 3 and 4, Table 5). The di-substituted substrate, 2,4-dichlorobenzyl alcohol, gives much higher yields of the

reaction product (2,4-dichlorobenzaldehyde) than the p-mono-substituted 4-chlorobenzyl alcohol under the same conditions (entries 2 and 6, Table 5) [50].

**Table 4.** Catalytic results (yield and selectivity for aldehyde) for the oxidation of benzyl alcohols using $Fe_3O_4@SiO_2@VO(ephedrine)_2$ as catalyst [49] [a].

| Entry | Substrate | Product | Yield (%) | Selectivity (%) |
|---|---|---|---|---|
| 1 | | | 98 | >99 |
| 2 [b] | | | 91 | >99 |
| 3 [c] | | | 95 | >99 |
| 4 [d] | | | 96 | >99 |
| 5 [e] | | | 97 | >99 |
| 6 [f] | | | 98 | >99 |

[a] Reaction conditions: substrate (1 mmol), $Fe_3O_4@SiO_2@VO(ephedrine)_2$ (50 mg), polyethylene glycol (1 mL), TBHP 70% aq. sol. (1 mmol) as oxidant, 80 °C, 3 h; [b] TBHP 70% aq. sol. (4 mmol), 6 h; [c] TBHP 70% aq. sol. (4 mmol), 4.33 h; [d] 2 h; [e] 47 h; [f] TBHP 70% aq. sol. (4 mmol), 7 h.

**Table 5.** Yield and selectivity to aldehyde in the oxidation of various benzyl alcohols using $Fe_3O_4@MAPTMS@PAA@Triazole@Cu(I)$ as catalyst [50] [a].

| Entry | Substrate | Product | Yield (%) | Selectivity (%) |
|---|---|---|---|---|
| 1 [b] | | | 85 | >99 |
| 2 | | | 53 | >99 |
| 3 [c] | | | 67 | >99 |
| 4 [b] | | | 93 | >99 |
| 5 | | | 73 | >99 |
| 6 | | | 97 | >99 |

[a] Reaction conditions: substrate (2 mmol), $Fe_3O_4@MAPTMS@PAA@Triazole@Cu(I)$ (0.1 mol%), acetonitrile (3 mL), 60 °C, 2 h and $H_2O_2$ 30% aq. sol. (40 mmol) as oxidant; [b] 5 h; [c] 85 °C and 3 h.

Kong et al. tested the catalytic performance of a $Pd/Fe_3O_4@mCeO_2$ core–shell in the aerobic oxidation of benzyl alcohol to benzaldehyde at normal pressure. The multifunctional composite was prepared using a layer-by-layer method (Figure 4) [51]. Under optimized conditions (2.6 wt% Pd content, $CeO_2$ support, and temperature 100 °C), the core–shell nanostructured palladium catalyst was able to convert 80.5% of benzyl alcohol to benzaldehyde with 94.8% selectivity after 7 h (entry 1, Table 6). Being heterogeneous and magnetic, the catalyst was easily separated from the reaction mixture under an external magnetic field. This is a significant advantage of this system, as conventional separation techniques result in high catalyst losses. The catalyst could be reused and showed high efficiency even after seven catalytic cycles. Pd supported on $CeO_2$ or $TiO_2$ resulted in better catalytic activity compared to the $SiO_2$ or C supports. This effect can be explained by the

larger oxygen defects in the former supports, which increased the metal/support synergy and, at the same time activating $O_2$ more efficiently for the aerobic oxidation of benzyl alcohol. It was found that the introduction of an electron withdrawing group (Cl, $NO_2$) at the *para*-position of the alcohol molecule significantly decreased both the product yield and the reaction selectivity (entries 2 and 5, Table 6), whereas the introduction of electron donor groups (Me, MeO, $^{t}Bu$) did not significantly affect these parameters compared to the unsubstituted benzyl alcohol (entries 1, 3, 4, and 6, Table 6) [51]. However, the increased electronic density in the benzene ring of the aromatic alcohol favors its oxidation to the corresponding aldehyde [51].

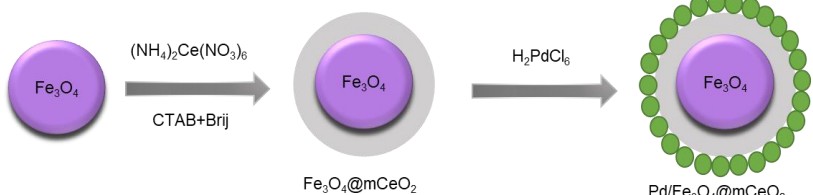

**Figure 4.** Preparation of the nanostructured $Pd/Fe_3O_4@mCeO_2$ core–shell (CTAB-cetyltrimethylammonium bromide; Brij—polyethylene glycol monododecyl ether). Adapted from [51] with permission from Springer Nature.

**Table 6.** Catalytic results (yield and selectivity for aldehyde) for the oxidation of benzyl alcohols using $Pd/Fe_3O_4@mCeO_2$ as catalyst [51] [a].

| Entry | Substrate | Product | Yield (%) | Selectivity (%) |
|:---:|:---:|:---:|:---:|:---:|
| 1 | | | 81 | 95 |
| 2 | | | 65 | 86 |
| 3 | | | 81 | 94 |
| 4 | | | 82 | 93 |
| 5 | | | 61 | 88 |
| 6 | | | 85 | 95 |

[a] Reaction conditions: substrate (5 mL), $Pd/Fe_3O_4@mCeO_2$ (substrate:metal molar ratio = 2500), 100 °C, 7 h under 1 atm $O_2$ (20 mL/min).

After the successful synthesis of the $Fe_3O_4@P4VP@FeCl_3$ core–shell by introducing the catalytically active $FeCl_3$ moiety into the $Fe_3O_4@P4VP$ (P4VP-poly(4-vinylpyridine)) core (4.36 wt% Fe content) through the interaction between P4VP and $FeCl_3$ (Figure 5), Li et al. investigated its ability as a catalyst for the aerobic oxidation of benzyl alcohol [52]. Reactions were carried out using 2 mol% (molar percentage based on iron) of $Fe_3O_4@P4VP@FeCl_3$, 0.1 mmol $NaNO_2$, 0.2 mmol of 2,2,6,6-tetramethylpiperidinyl-1-oxy (TEMPO), and 1 mmol of benzyl alcohol in acetonitrile for 12 h at 60 °C under 1 atm $O_2$. Using this catalytic system, it was possible to combine an excellent conversion with a high selectivity, both around 99% (entry 1, Table 7). In agreement with the previous study, the introduction of an electron withdrawing group (F) in the *para*-position of the aromatic ring was unfavorable for the oxidation, while the presence of electron donor substituents (Me, MeO) had no significant effect on the yield and selectivity (entries 2–4, Table 7). It should also be noted that reactions carried out in the presence of $Fe_3O_4$ and $Fe_3O_4@P4VP$ failed, demonstrating that the $FeCl_3$ moiety was an active part of the catalyst [52].

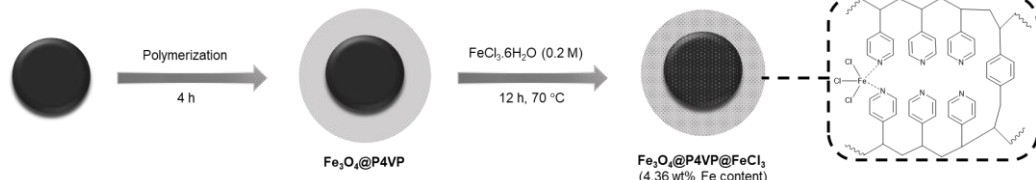

**Figure 5.** Preparation of Fe₃O₄@P4VP@FeCl₃ core–shell.

**Table 7.** Catalytic results (yield and selectivity to aldehyde) in the oxidation of benzyl alcohols using Fe₃O₄@P4VP@FeCl₃ as catalyst [52] [a].

| Entry | Substrate | Product | Yield (%) | Selectivity (%) |
|:---:|:---:|:---:|:---:|:---:|
| 1 | benzyl alcohol | benzaldehyde | 99 | 99 |
| 2 | 4-fluorobenzyl alcohol | 4-fluorobenzaldehyde | 80 | 81 |
| 3 | 4-methylbenzyl alcohol | 4-methylbenzaldehyde | 99 | 99 |
| 4 | 4-methoxybenzyl alcohol | 4-methoxybenzaldehyde | 95 | 99 |

[a] Reaction conditions: substrate (1 mmol), Fe₃O₄@P4VP@FeCl₃ (2 mol%), acetonitrile (5 mL), TEMPO (0.2 mmol), KNO₂ (0.2 mmol), 60 °C, 12 h under 1 atm O₂.

Lang et al. synthesized the structured spherical nanocomposite consisting of a core of γ-Fe₂O₃ and a shell of mesoporous silica and applied it as a catalyst in the oxidation of benzyl alcohols. The unsubstituted benzyl alcohol showed the best performance in this reaction with a conversion and selectivity of more than 90% (entry 1, Table 8). The substituted benzyl alcohols either with the donor group Me or with the acceptor group NO₂ give much lower product yields, but the conversion of *p*-nitrobenzyl alcohol is very selective (98%, entries 2 and 3, Table 8). In the recycling experiments, the catalyst showed some loss of activity after three cycles, but the selectivity remained above 80% [53].

**Table 8.** Catalytic results (yield and selectivity to aldehyde) in the oxidation of benzyl alcohols using γ-Fe₂O₃@m-SiO₂ as catalyst [53] [a].

| Entry | Substrate | Product | Yield (%) | Selectivity (%) |
|:---:|:---:|:---:|:---:|:---:|
| 1 | benzyl alcohol | benzaldehyde | 94 | 92 |
| 2 | 4-methylbenzyl alcohol | 4-methylbenzaldehyde | 60 | 84 |
| 3 | 4-nitrobenzyl alcohol | 4-nitrobenzaldehyde | 71 | 98 |

[a] Reaction conditions: substrate (10 mmol), γ-Fe₂O₃@m-SiO₂ (50 mg), polyethylene glycol (10 mL), 130 °C, 24 h under O₂.

Qin et al. synthesized the hollow cubic Au@Zn/Ni-MOF-2 core–shell nanoreactor and applied it to different substrates, namely benzyl alcohol and derivatives (Table 9), in toluene at 95 °C and using air as the green oxidant. This core–shell showed excellent catalytic performance towards benzyl alcohol after 2 h of reaction (entry 1, Table 9). The catalyst was reused after recovery in five cycles without loss of catalytic activity (yield and selectivity). The improved stability of the hollow core–shell Au@Zn/Ni-MOF-2 compared to pure AuNPs highlights the synergistic core–shell effect. Substituted alcohols have a significantly lower activity towards oxidation. For the *o*-nitrobenzyl alcohol, where the substituent is close to the oxidizing hydroxylic group, a product yield of only 8% was

obtained. Such an inactivation may be due to steric hindrance imposed by the *ortho* NO$_2$ substituent (entries 2–4, Table 9) [54].

**Table 9.** Catalytic results (yield and selectivity to aldehyde) for the oxidation of benzyl alcohols using Au@Zn/Ni-MOF-2 as catalyst [54] [a].

| Entry | Substrate | Product | Yield (%) | Selectivity (%) |
|---|---|---|---|---|
| 1 | (benzyl alcohol) | (benzaldehyde) | 98 | 99 |
| 2 | (4-methylbenzyl alcohol) | (4-methylbenzaldehyde) | 46 | 99 |
| 3 | (2-nitrobenzyl alcohol) | (2-nitrobenzaldehyde) | 8 | 99 |
| 4 | (3-nitrobenzyl alcohol) | (3-nitrobenzaldehyde) | 51 | 99 |

[a] Reaction conditions: substrate (0.2 mmol), Au@Zn/Ni-MOF-2 (15 mg), toluene (6 mL), 95 °C, 2 h under O$_2$ atmosphere (atm).

In 2022, Ardakani et al. immobilized the dioxo-molybdenum (VI) complex with unsymmetrical Schiff base on the CoFe$_2$O$_4$@SiO$_2$ core–shell and named it CoFe$_2$O$_4$@SiO$_2$ @[MoO$_2$(salenac-OH)] (where salenac-OH is [9-(2′,4′-dihydroxyphenyl)-5,8-diaza-4-methylnone-2,4,8-trienato)](-2)). This composite was used as a catalyst in various peroxidative oxidations of alcohols (TBHP, 70% aq. sol.). The reaction conditions, including the type of solvent, the amount of catalyst, and the type and amount of oxidant, were optimized for the benzyl alcohol. A yield of 90%, a selectivity of 98%, and a TON of 66.7 were recorded after 2 h (entry 1, Table 10). Due to the magnetic properties of the catalyst, it was removed using an external magnet. A reduction of about 10% in the reaction yield was observed after four cycles of the catalyst recovery. This effect may be related to the leaching of Mo. The substrates with the electron donor substituents showed higher activity (entries 2–6, Table 10) [55].

**Table 10.** Catalytic results (yield and selectivity for aldehyde) for the oxidation of benzyl alcohols using CoFe$_2$O$_4$@SiO$_2$@[MoO$_2$(salenac-OH)] as catalyst [55] [a].

| Entry | Substrate | Product | Yield (%) | Selectivity (%) |
|---|---|---|---|---|
| 1 | (benzyl alcohol) | (benzaldehyde) | 90 | 98 |
| 2 | (4-chlorobenzyl alcohol) | (4-chlorobenzaldehyde) | 80 | 93 |
| 3 | (4-methylbenzyl alcohol) | (4-methylbenzaldehyde) | 95 | 98 |
| 4 | (4-methoxybenzyl alcohol) | (4-methoxybenzaldehyde) | 93 | 97 |
| 5 | (4-nitrobenzyl alcohol) | (4-nitrobenzaldehyde) | 73 | 95 |
| 6 | (4-tert-butylbenzyl alcohol) | (4-tert-butylbenzaldehyde) | 97 | 95 |

[a] Reaction conditions: substrate (1 mmol), CoFe$_2$O$_4$@SiO$_2$@[MoO$_2$(salenac-OH)] (50 mg), r.t., 2 h and TBHP (2 mmol).

In 2023, Hou et al. applied the Fe$_3$O$_4$@Cu$_3$(BTC)$_2$ core–shell in the aerobic oxidation of benzyl alcohols studying the effect of different substituents. This composite shows a higher catalytic activity in the presence of electron donating groups (MeO) compared to

the electron withdrawing groups (F) (entries 2 and 3, Table 11). This material also shows excellent catalytic activity for the unsubstituted benzyl alcohol (entry 1, Table 11) [56].

**Table 11.** Catalytic results (yield and selectivity for aldehyde) for the oxidation of benzyl alcohols using $Fe_3O_4@Cu_3(BTC)_2$ as catalyst [56] [a].

| Entry | Substrate | Product | Yield (%) | Selectivity (%) |
|---|---|---|---|---|
| 1 | | | 99 | 99 |
| 2 | | | 78 | 99 |
| 3 | | | 95 | 99 |

[a] Reaction conditions: substrate (0.5 mmol), $Fe_3O_4@Cu_3(BTC)_2$ (1% molar fraction based on Cu), $CH_3CN$ (2.5 mL), $NaHCO_3$ (0.5 mmol), TEMPO (0.25 mmol), 60 °C and 12 h under 1 atm $O_2$.

Recently, Dabiri et al. synthesized the $Fe_3O_4@PDA/Pd@N$-RGO core–shell which consists of a magnetite core coated with polydopamine (PDA), to which palladium nanoparticles are subsequently immobilized to this set. This structured composite is surrounded by a nitrogen-doped reduced graphene oxide (N-RGO) layer. The catalytic activity of this material was investigated in the aerobic oxidation of alcohols. Benzyl alcohol was used to optimize parameters such as solvent type, the base, the Pd content, and temperature. The yield of 94% and the selectivity towards benzaldehyde of 99% were obtained for the optimized conditions (entry 1, Table 12). The catalyst was stable after five recovery cycles, with no significant losses, but the yield decreased from 94% to 87%. The effect of different substituents was also investigated. The substrates with the electron acceptor groups (F and $NO_2$) showed lower conversions, although the *p*-chlorobenzylic alcohol was as active as the unsubstituted substrate and the alcohols with the electron donor groups (Table 12) [57].

**Table 12.** Catalytic results (yield and selectivity for aldehyde) for the oxidation of benzyl alcohols using $Fe_3O_4@PDA/Pd@N$-RGO as catalyst [57] [a].

| Entry | Substrate | Product | Yield (%) | Selectivity (%) |
|---|---|---|---|---|
| 1 | | | 94 | 99 |
| 2 | | | 93 | 99 |
| 3 | | | 95 | 99 |
| 4 | | | 90 | 99 |
| 5 | | | 81 | 99 |
| 6 | | | 86 | 99 |

[a] Reaction conditions: substrate (1 mmol), $Fe_3O_4@PDA/Pd@N$-RGO (0.05% molar fraction based on Pd), toluene (3 mL), $K_2CO_3$ (1 mmol), 90 °C, and 2.5 h under 1 atm $O_2$.

### 2.2. Phenylethanols

Yin et al. studied the aerobic oxidation of 1-phenylethanol to acetophenone (Scheme 2) catalyzed by a series of 3D hierarchical hollow nanostructured magnetic catalysts $\gamma$-$Fe_2O_3@M_3Al$-LDH@$Au_{25}$-x (M = Ni, Mg or Cu/Mg (0.5/2.5); LDH = layered double hydroxide; x = Au loading in wt%). The synergistic interaction of different supports with the $Au_{25}$ nanoclusters ($Au_{25}$NCs) as well as the effect of the gold content were investigated.

The lower gold-loaded catalyst $\gamma$-Fe$_2$O$_3$@Ni$_3$Al-LDH@Au$_{25}$-0.053 showed the best performance (TOF: 112,498 h$^{-1}$) with excellent conversion and selectivity, both 99% (entry 1, Table 13). This can be attributed to the near atomic precision of the electron-rich Au$_{25}$NCs magnetic core, as well as the three-phase synergetic effect of the Au$_{25}$NCs = Ni$_3$Al-LDH magnetic core enhanced by abundant Ni-OH sites [37]. The Fe$_3$O$_4$@Ni$_3$Al-LDH support and the Fe$_3$O$_4$ core were also tested, but both showed low conversion towards 1-phenylethanol in both cases. This confirms that the gold clusters are the active sites for the alcohol oxidation.

**Scheme 2.** Oxidation of 1-phenylethanol to acetophenone catalyzed by the Fe$_2$O$_3$@Ni$_3$Al-LDH@Au$_{25}$-0.053 nanostructure.

**Table 13.** Yield and selectivity in the oxidation of 1-phenylethanol using different core–shell as the catalysts.

| Entry | Catalyst | Yield (%) | Selectivity (%) | Ref. |
|---|---|---|---|---|
| 1 [a] | $\gamma$-Fe$_2$O$_3$@Ni$_3$Al-LDH@Au$_{25}$-0.053 | 99 | 99 | [37] |
| 2 [b] | Fe$_3$O$_4$@P4VP@FeCl$_3$ | 50 | 51 | [52] |
| 3 [c] | Fe$_3$O$_4$@SiO$_2$@VO(ephedrine)$_2$ | 96 | >99 | [49] |
| 4 [d] | Fe$_3$O$_4$@TiO$_2$@[FeCl$_2${$\kappa^3$-HC(pz)$_3$}] | 97 | 67 | [58] |
| 5 [e] | Fe$_3$O$_4$@PDA/Pd@N-RGO | 82 | 99 | [57] |
| 6 [f] | Fe$_3$O$_4$@Cu$_3$(BTC)$_2$ | 57 | 99 | [56] |
| 7 [g] | CoFe$_2$O$_4$@SiO$_2$@[MoO$_2$(salenac-OH)] | 75 | 100 | [55] |

[a] Reaction conditions: 1-phenylethanol (1 mmol), $\gamma$-Fe$_2$O$_3$@Ni$_3$Al-LDH@Au$_{25}$-0.053 (Au: 0.115%), toluene (5 mL), 80 °C, 0.5 h under 1 atm O$_2$ (20 mL/min); [b] 1-phenylethanol (1 mmol), 2 mol% of Fe$_3$O$_4$@P4VP@FeCl$_3$, acetonitrile (5 mL), TEMPO (0.2 mmol), KNO$_2$ (0.2 mmol), 60 °C, 12 h under 1 atm O$_2$; [c] 1-phenylethanol (1 mmol), Fe$_3$O$_4$@SiO$_2$@VO(ephedrine)$_2$ (50 mg), polyethylene glycol (1 mL), 80 °C, 3.25 h and TBHP 70% aq. sol. (4 mmol) as oxidant; [d] 1-phenylethanol (5 mmol), Fe$_3$O$_4$@TiO$_2$@[FeCl$_2${$\kappa^3$-HC(pz)$_3$}] (1.56 µmol), 80 °C, 3 h and TBHP 70% aq. sol. (10 mmol) as oxidant; [e] 1-phenylethanol (1 mmol), Fe$_3$O$_4$@PDA/Pd@N-RGO (Pd: 0.05 mol%), toluene (3 mL), K$_2$CO$_3$ (1 mmol), 90 °C, 2.5 h under 1 atm O$_2$; [f] 1-phenylethanol (0.5 mmol), Fe$_3$O$_4$@Cu$_3$(BTC)$_2$ (Cu: 1 mol%), acetonitrile (2.5 mL), NaHCO$_3$ (0.5 mmol), TEMPO (0.25 mmol), 60 °C, 12 h under 1 atm O$_2$; [g] 1-phenylethanol (1 mmol), CoFe$_2$O$_4$@SiO$_2$@[MoO$_2$(salenac-OH)] (50 mg), r.t., 2.5 h and TBHP 70% aq. sol. (2 mmol).

This core–shell was easily recovered from the reaction mixture due to its magnetic properties and then reused for 10 cycles without any significant loss of activity, indicating the high stability of the catalyst and revealing Fe$_3$O$_4$@LDH as a promising green platform to support other metal nanoparticles [37].

In 2017, Li et al. successfully synthesized the core–shell Fe$_3$O$_4$@P4VP@FeCl$_3$ and applied it to several aerobic oxidation reactions of different alcohols. They found that the core–shell does not play as much of an active role in secondary alcohols, such as 1-phenylethanol (entry 2, Table 13), as in primary alcohols, such as benzyl alcohol (99% yield and selectivity, entry 1, Table 7) [52].

The peroxidative oxidation of 1-phenylethanol (TBHP 70% aq. sol.) in the presence of the recyclable magnetic nanoparticle-supported oxo-vanadium ephedrine complex Fe$_3$O$_4$@SiO$_2$@VO(ephedrine)$_2$ was first reported by Rostami et al. in 2017. Acetophenone was the only reaction product reaching 96% yield after approx. 3 h (entry 3, Table 13) [49].

More recently, Matias et al. reported the oxidation of 1-phenyletanol in the presence of a C-scorpionate iron(II) complex supported on the magnetic core–shell Fe$_3$O$_4$@TiO$_2$@[FeCl$_2${$\kappa^3$-HC(pz)$_3$}] under unconventional energy inputs (mechanical, thermal, sonication, and microwave irradiation). This hybrid material showed high catalytic activity achieving the 97% conversion of 1-phenylethanol into acetophenone, whereas the Fe$_3$O$_4$/TiO$_2$ core–shell did not go beyond 4% under the same reaction conditions (3 h, 40 °C) (entry 4, Table 13). Despite the high conversion rate, acetophenone was not the only product

detected. The other products, styrene and benzaldehyde, resulted from dehydration of the alcohol and subsequent oxidation [58].

In 2022, Dabiri et al. extended the application of the $Fe_3O_4$@PDA/Pd@N-RGO core–shell beyond benzyl alcohols and tested it as a catalyst for the aerobic oxidation reaction of 1-phenylethanol to acetophenone with a good yield of 85%, although lower compared to the oxidation of the benzyl alcohols (entry 5, Table 13) [57].

More recently, Hou et al. aerobically oxidized 1-phenylethanol using the core–shell $Fe_3O_4$@$Cu_3(BTC)_2$ as the catalyst. A yield of 57% acetophenone was obtained after 12 h (Entry 6, Table 13) [56]. Ardakani et al. tested the catalytic activity of the composite $CoFe_2O_4$@$SiO_2$@[$MoO_2$(salenac-OH)] in the same reaction using ${}^tBuOOH$ as oxidant (acetophenone yield of 75%, entry 7, Table 13). Room temperature and solvent-free conditions were used [55].

Rostami et al. extended the application of the $Fe_3O_4$@$SiO_2$@VO(ephedrine)$_2$ core–shell to the peroxidative oxidation of the 2-phenylethanol isomer in polyethylene glycol (PEG) as a green solvent (Scheme 3) under optimal reaction conditions, achieving a 98% isolated yield of the corresponding carbonyl compound but after a longer reaction time (10 h) (Entry 1, Table 14) [49].

**Scheme 3.** Selective oxidation of 2-phenylethanol to 2-phenylacetaldehyde catalyzed by $Fe_3O_4$@$SiO_2$@VO(ephedrine)$_2$.

**Table 14.** Yield and selectivity in the oxidation of 2-phenylethanol using different core–shell as the catalysts.

| Entry | Catalyst | Yield (%) | Selectivity (%) | Ref. |
|-------|----------|-----------|-----------------|------|
| 1 [a] | $Fe_3O_4$@$SiO_2$@VO(ephedrine)$_2$ | 98 | >99 | [49] |
| 2 [b] | Au@Zn/Ni-MOF-2 | 70 | 99 | [54] |

[a] Reaction conditions: 2-phenylethanol (1 mmol), $Fe_3O_4$@$SiO_2$@VO(ephedrine)$_2$ (50 mg), polyethylene glycol (1 mL) as solvent, 80 °C, 10 h and TBHP 70% aq. sol. (4 mmol) as oxidant; [b] 2-phenylethanol (0.2 mmol), Au@Zn/Ni-MOF-2 (15 mg), toluene (6 mL), 95 °C, 2 h under $O_2$ atmosphere (1 atm).

In 2021, the application of the hollow bimetallic organic framework core–shell, Au@Zn/Ni-MOF-2, by Qin et al. in the aerobic oxidation of 2-phenylethanol in toluene at 95 °C and using $O_2$ as the green oxidant and after 2 h of reaction gave a yield of 70% to 2-phenylacetaldehyde [54].

## 2.3. Benzhydrol

In 2016, Yin et al. reported the synthesis of the hollow nanostructured supported gold nanoclusters and revealed the $\gamma$-$Fe_2O_3$@$Ni_3$Al-LDH@$Au_{25}$-0.053 core–shell as a highly efficient catalyst for the conversion of benzhydrol to benzophenone (Scheme 4). Among other reported potential catalysts (Table 15), this core–shell composite appears to be the most selective catalyst for the benzhydrol oxidation (entry 1, Table 15) [37].

**Scheme 4.** Oxidation of benzhydrol to benzophenone.

**Table 15.** Catalytic results (yield and selectivity to ketone) for the oxidation of benzhydrol using different core–shell as catalysts.

| Entry | Catalyst | Yield (%) | Selectivity (%) | Ref. |
|-------|----------|-----------|-----------------|------|
| 1 [a] | $\gamma$-Fe$_2$O$_3$@Ni$_3$Al-LDH@Au$_{25}$-0.053 | 99 | 99 | [37] |
| 2 [b] | Fe$_3$O$_4$@SiO$_2$@VO(ephedrine)$_2$ | 94 | >99 | [49] |
| 3 [c] | Fe$_3$O$_4$@MAPTMS@PAA@Triazole@Cu(I) | 43 | >99 | [50] |
| 4 [d] | RuO$_2$@ZrO$_2$ | 85 | 94 | [48] |

[a] Reaction conditions: benzhydrol (1 mmol), $\gamma$-Fe$_2$O$_3$@Ni$_3$Al-LDH@Au$_{25}$-0.053 (Au: 0.115%), toluene (5 mL), 80 °C, 3 h under 1 atm O$_2$ (20 mL/min); [b] benzhydrol (1 mmol), Fe$_3$O$_4$@SiO$_2$@VO(ephedrine)$_2$ (50 mg), polyethylene glycol (1 mL), 80 °C, 4.83 h and TBHP 70% aq. sol. (4 mmol) as oxidant; [c] benzhydrol (2 mmol), Fe$_3$O$_4$@MAPTMS@PAA@Triazole@Cu(I) (0.1 mol%), acetonitrile (3 mL), 80 °C, 7 h and H$_2$O$_2$ 30% aq. sol. (40 mmol) as oxidant; [d] benzhydrol (1 mmol), RuO$_2$@ZrO$_2$ (20 mg), acetonitrile (5 mL), 80 °C, 0.08 h (5 min) and TBHP 70% aq. sol. (0.3 mL) as oxidant.

Promising results obtained by Rostami et al. in the peroxidative oxidation (TBHP 70% aq. sol.) of benzhydrol (entry 2, Table 15) using the magnetic Fe$_3$O$_4$@SiO$_2$@VO(ephedrine)$_2$ core–shell catalyst [49] should be highlighted, although this reaction requires a slightly longer reaction time compared to $\gamma$-Fe$_2$O$_3$@Ni$_3$Al-LDH@Au$_{25}$-0.053 (entry 1, Table 15) [37].

A rather moderate activity (in terms of the product yield and reaction time) was reported by Ghalavand et al. for the oxidation of benzhydrol catalyzed by the structured nanocomposite Fe$_3$O$_4$@MAPTMS@PAA@Triazole@Cu(I) core–shell and using H$_2$O$_2$ as an environmentally friendly oxidant (entry 3, Table 15). In fact, such a relatively low yield could be attributed to the possible degradation of the oxidant under the applied reaction conditions, namely high temperature, rather than to the catalyst activity itself. Nevertheless, the Fe$_3$O$_4$@MAPTMS@PAA@Triazole@Cu(I) core–shell showed complete selectivity towards benzophenone [50].

It is worth mentioning the excellent results reported by Shojaei et al. for the peroxidative oxidation (TBHP 70% aq. sol.) of benzhydrol with the core–shell catalyst RuO$_2$@ZrO$_2$ achieving a conversion of 85% in a very short time (5 min) (entry 4, Table 15) [48]. After verifying its activity for benzhydrol oxidation, Shojaei et al. applied the RuO$_2$@ZrO$_2$ core–shell for the oxidation of the benzhydrol chloro derivative, 4-chlorobenzhydrol (Scheme 5). The yield and selectivity towards the reaction product (4-chlorobenzophenone) reached 95.0 and 94.7%, respectively [48].

**Scheme 5.** Oxidation of 4-chlorobenzhydrol to 4-chlorobenzophenone catalyzed by RuO$_2$@ZrO$_2$ core–shell.

### 2.4. Cinnamyl Alcohol

In 2016, Yin et al. demonstrated the great versatility of the $\gamma$-Fe$_2$O$_3$@Ni$_3$Al-LDH@Au$_{25}$ composite extending its application as catalyst to a wide range of alcohols. The almost atomically precise gold catalyst showed high yield and selectivity not only for benzylic secondary alcohols and aliphatic secondary alcohols such as 1-phenylethanol and benzhydrol as mentioned above but also for allylic alcohols, i.e., cinnamyl alcohol (Scheme 6). The hierarchical hollow core@shell magnetic nanogold catalyst selectively oxidizes cinnamyl alcohol to cinnamaldehyde with the 95% conversion after 4 h in toluene at 80 °C and using 1 atm O$_2$ (20 mL/min) (entry 1, Table 16) [37].

Similar yields and selectivities were obtained by Li et al. for the Fe$_3$O$_4$@P4VP@FeCl$_3$ nanospheres in the aerobic oxidation of cinnamyl alcohol to cinnamaldehyde, although a longer period of 12 h was required to achieve these values (entry 2, Table 16) [52].

Cinnamaldehyde (selectivity >99%) was obtained in good yield (75%) starting from the corresponding allylic alcohol (entry 3, Table 16) in the presence of the core–shell $Fe_3O_4$@MAPTMS@PAA@Triazole@Cu(I) nanocomposite as reported by Ghalavand et al. [50].

In 2020, Zhao et al. tested the catalytic activity of Ru@quinone core–shell in the oxidation of cinnamyl alcohol. Excellent conversion and selectivity were achieved for the 50 mg loading of the catalyst in the presence of $O_2$ as an oxidant (entry 4, Table 16) [47].

**Scheme 6.** Oxidation of cinnamyl alcohol to cinnamaldehyde catalyzed by the $\gamma$-$Fe_2O_3$@$Ni_3$Al-LDH@$Au_{25}$ composite.

**Table 16.** Catalytic results (yield and selectivity to aldehyde) for the oxidation of cinnamyl alcohol using different core–shell as catalyst.

| Entry | Catalyst | Yield (%) | Selectivity (%) | Ref. |
|---|---|---|---|---|
| 1 [a] | $\gamma$-$Fe_2O_3$@$Ni_3$Al-LDH@$Au_{25}$-0.053 | 95 | 93 | [37] |
| 2 [b] | $Fe_3O_4$@P4VP@$FeCl_3$ | 94 | 95 | [52] |
| 3 [c] | $Fe_3O_4$@MAPTMS@PAA@Triazole@Cu(I) | 75 | >99 | [50] |
| 4 [d] | Ru@SQ | 90 | 95 | [47] |

[a] Reaction conditions: cinnamyl alcohol (1 mmol), $\gamma$-$Fe_2O_3$@$Ni_3$Al-LDH@$Au_{25}$-0.053 (Au: 0.115%), toluene (5 mL), 80 °C, 4 h under 1 atm $O_2$ (20 mL/min); [b] cinnamyl alcohol (1 mmol), 2 mol% of $Fe_3O_4$@P4VP@$FeCl_3$ catalyst, acetonitrile (5 mL), TEMPO (0.2 mmol), $KNO_2$ (0.2 mmol), 60 °C, 12 h under 1 atm $O_2$; [c] cinnamyl alcohol (2 mmol), $Fe_3O_4$@MAPTMS@PAA@Triazole@Cu(I) (0.1 mol%), acetonitrile (3 mL), 85 °C, 2 h and $H_2O_2$ 30% aq. sol. (40 mmol) as oxidant; [d] cinnamyl alcohol (7.5 mmol), Ru@SQ (50 mg), 100 °C, 18 h under 1 atm $O_2$.

## 2.5. Other Alcohols

Yang et al. synthesized hexagonal prisms of Co(II/III)@ZnO contemplated by MOF and successfully applied them to the oxidation of vanillic alcohol to vanillin [59]. The imidazo-late zeolitic framework material ZIF-67 was used as a core, and the ZnO shell was introduced using a simple two-step method, seed growth and calcination under inert atmosphere. The Co(II/III)@ZnO structured composites with core and shell act as efficient and biocompatible catalysts for the selective oxidation of vanillic alcohol to vanillin, with a conversion of 20% vanillic alcohol and a selectivity of 80% vanillin after 2 h of reaction. The prismatic structure is responsible for the easy separation of the catalyst allowing its reuse in three cycles without significant modification of the catalyst structure.

Rostami et al. oxidized 1-indanol to 1-indanone applying the $Fe_3O_4$@$SiO_2$ core–shell supported on VO(ephedrine)$_2$ as a catalyst. The product yield of 93% was achieved after 7 h of reaction with *tert*-butyl hydroperoxide (TBHP) used as an oxidant (entry 2, Table 17) [49].

**Table 17.** Catalytic results (yield and selectivity to aldehyde) for the oxidation of other alcohols using different core–shell as catalysts.

| Entry | Catalyst | Substrate | Product | Yield (%) | Selectivity (%) | Ref. |
|---|---|---|---|---|---|---|
| 1 [a] | Co(II/III)@ZnO |  |  | 20 | 81 | [59] |
| 2 [b] | $Fe_3O_4$@$SiO_2$ @VO(ephedrine)$_2$ |  |  | 93 | >99 | [49] |
| 3 [c] | Au@Zn/Ni-MOF-2 |  |  | 77 | 99 | [54] |

[a] Reaction conditions: vanillyl alcohol (3.25 mmol), Co(II/III)@ZnO (100 mg), acetonitrile (90 mL), 140 °C, 2 h and 3 MPa $O_2$; [b] 1-indanol (1 mmol), $Fe_3O_4$@$SiO_2$@VO(ephedrine)$_2$ (50 mg), polyethylene glycol (1 mL), 80 °C, 7.25 h and TBHP 70% aq. sol. as oxidant (4 mmol); [c] 1-naphthalenemethanol (0.2 mmol), Au@Zn/Ni-MOF-2 (15 mg), toluene (6 mL), 95 °C, 2 h under 1 atm $O_2$.

In 2021, Qin et al. selectively oxidized 1-naphthalenemethanol in air to the corresponding aldehyde with a hollow core–shell Au@Zn/Ni-MOF-2 and obtained a remarkable conversion (77.0%, entry 3 of Table 17) when compared to the gold nanoparticles which recorded only an 8.0% yield [54].

## 3. Oxidation of Aromatic Heterocyclic Alcohols

### 3.1. Furfuryl Alcohol

As with benzyl alcohol (a primary alcohol), the oxidation of furfuryl alcohol (Scheme 7) can lead to two further oxidized products, i.e., furfural and the corresponding acid.

**Scheme 7.** Oxidation of furfuryl alcohol to furfural and furoic acid catalyzed by the Fe$_3$O$_4$@SiO$_2$@VO(ephedrine)$_2$ composite.

Rostami et al. tested the catalytic activity of the Fe$_3$O$_4$@SiO$_2$@VO(ephedrine)$_2$ composite with the supported vanadium complex in the oxidation of furfuryl alcohol (1 mmol) using *tert*-butyl hydroperoxide (1 mmol) as an oxidant and PEG as a solvent at 80 °C for 50 h yielding exclusively furfural with 85% conversion of the alcohol [49]. More recently, Zhao et al. achieved a similar yield (80%) of furfural in the oxidation of furfuryl alcohol (0.5 g) after only 3 h using 50 mg of the structured ruthenium nanocomposite wrapped in disodium anthraquinone-2,6-disulfonate as catalyst and O$_2$ as an oxidant at 90 °C. The selectivity of the reaction towards furfural was 92% [47].

### 3.2. 2,5-Hydroxymethylfurfural

The oxidation of 2,5-hydroxymethylfurfural is more difficult to control than the oxidation of furfuryl alcohol because the former has two functional groups that are susceptible to oxidation. As a result, four oxidation products can be obtained (Scheme 8).

**Scheme 8.** Oxidation of 2,5-hydroxymethylfurfural (HMF) to 2,5-diformylfuran (DFF), 5-hydroxymethyl-furan-2-carboxylic acid (HMFCA), 5-formyl-2-furoic acid (FFCA), and 2,5-furandicarboxylic acid (FDCA).

In 2020, Zhao et al. used the Ru@SQ nanocomposite (ruthenium nanoparticles coated with disodium anthraquinone-2,6-disulfonate) as a catalyst for the oxidation of 2,5-hydroxymethylfurfural (O$_2$ as an oxidant, 50 mg of catalyst, 0.4 g of the substrate at 110 °C, 16 h) with a conversion of 69% and with an 89% selectivity towards 2,5-diformylfuran [47].

More recently, Song et al. oxidized the aldehyde group of 2,5-hydroxymethylfurfural to give 5-hydroxymethyl-furan-2-carboxylic acid as the sole product. The D-CeO$_2$@N/C@TiO$_2$ core-shell with the CeO$_2$ spherical core (diameter of ca. 1600 nm in) coated with the *N*-doped carbon and TiO$_2$ was used as the catalyst. Under optimized reaction conditions (buffer solution of Na$_2$B$_4$O$_7$ with pH 9.18, 1 MPa O$_2$, 0.1 g of HMF, 100 mg of catalyst at

80 °C for 30 min), a conversion of 87.8% was achieved. The results were reproducible over four catalytic cycles [60].

### 3.3. Pyridinemethanol

In 2017, Li et al. tested the $Fe_3O_4$@P4VP@$FeCl_3$ core–shell as catalyst in the conversion of 2-pyridinemethanol at 60 °C into the corresponding aldehyde (entry 1, Table 18) [52].

**Table 18.** Catalytic results (yield and selectivity) for the oxidation of pyridinemethanol using different core–shell as catalysts.

| Entry | Catalyst | Substrate | Main Product | Yield (%) | Selectivity (%) | Ref. |
|---|---|---|---|---|---|---|
| 1 [a] | $Fe_3O_4$@P4VP@$FeCl_3$ | | | 20 | 81 | [52] |
| 2 [b] | $Fe_3O_4$@$SiO_2$ @VO(ephedrine)$_2$ | | | 93 | >99 | [49] |
| 3 [c] | Au@Zn/Ni-MOF-2 | | | 37 | 99 | [54] |

[a] Reaction conditions: 2-pyridinemethanol (1 mmol), 2 mol% of $Fe_3O_4$@P4VP@$FeCl_3$ catalyst, acetonitrile (5 mL), TEMPO (0.2 mmol), $KNO_2$ (0.2 mmol), 60 °C, 12 h under 1 atm $O_2$; [b] 3-pyridinemethanol (1 mmol), $Fe_3O_4$@$SiO_2$@VO(ephedrine)$_2$ (50 mg), polyethylene glycol (1 mL), 80 °C, 15 h and TBHP 70% aq. sol. (4 mmol) as oxidant; [c] 2,6-pyridinedimethanol (0.2 mmol), Au@Zn/Ni-MOF-2 (15 mg), toluene (6 mL), 95 °C, 2 h, under 1 atm $O_2$.

In the same year, Rostami et al. selectively oxidized another isomer of this alcohol (3-pyridinemethanol) to 3-pyridinecarboxaldehyde with $Fe_3O_4$@$SiO_2$@VO(ephedrine)$_2$ as a catalyst and TBHP 70% aq. sol. as an oxidant at 80 °C and achieved the yield of 93% after 15 h (entry 2, Table 18) [49].

Finally, Qin et al. recently tested Au@Zn/Ni-MOF-2 in the aerobic oxidation of pyridine-based diol. 2,6-Pyridinedicarboxaldehyde was quantitatively obtained with the yield of 37% after 2 h at 95 °C [54].

## 4. Oxidation of Aliphatic Alcohols

### 4.1. 1,2-Propanediol and Glycerol ($C_3$)

Cheong and co-workers investigated the effect of core–shell shape and composition on the catalytic performance of different Au@Pd core–shell in the oxidation of glycerol and propane-1,2-diol to glyceric acid and lactic acid, respectively. They found that the highly faceted icosahedral structures of the catalyst improved the catalytic conversion of these substrates compared to those surrounded by multiple facets (from 13.5% to 43% for glycerol and from 12.6 to 53.5% for 1,2-propanediol) [61]. Moreover, the monometallic nanoparticles were found to be less active than the mixed metal ones, converting 17.3 and 3.2% of 1,2-propanediol in the presence of PdNPs and AuNPs, respectively, and 25.8 and 8.5% of glycerol in the presence of PdNPs and AuNPs, respectively. All these catalysts are highly selective, achieving selectivities of 85–88% for glyceric acid and 77–85% for lactic acid [61]. The difference in conversion between the Au-based NPs and the Pd-based NPs could be attributed to a higher resistance to over-oxidation exhibited by the Au catalysts compared to the Pd catalysts under the applied oxidation conditions [61].

### 4.2. Crotyl Alcohol ($C_4$)

Lee et al. used titania-supported Au shell–Pd bimetallic nanoparticles Pd@Au@$TiO_2$ as catalysts for the aerobic oxidation of crotyl alcohol (Scheme 9) and obtained ca. 85% selectivity towards crotonaldehyde at 60 °C for 24 h under air using only 50 mg of catalyst and toluene as a solvent (entry 3, Table 19) [62].

**Scheme 9.** Oxidation of crotyl alcohol to crotonaldehyde and crotonic acid.

**Table 19.** Catalytic results (yield and selectivity) for the oxidation of various aliphatic alcohols using different core–shell as catalysts.

| Entry | Catalyst | Substrate | Product | Yield (%) | Selectivity (%) | Ref. |
|---|---|---|---|---|---|---|
| 1 [a] | Au@Pd | | | 54 | 88 | [61] |
| 2 [a] | Au@Pd | | | 43 | 85 | [61] |
| 3 [b] | Pd@Au@TiO$_2$ | | | 45 | 85 | [62] |
| 4 [c] | Au@Pd | | | 71 | 88 | [63] |
| 5 [d] | Ru@SQ | | | 57 | 74 | [47] |
| 6 [e] | Ru@SQ | | | 90 | 95 | [47] |
| 7 [f] | Au@Zn /Ni-MOF-2 | | | 33 | 99 | [54] |

[a] Reaction conditions: substrate (0.3 M), NaOH (2 eq. mol), catalyst (substrate/metal ratio 1000:1), 60 °C, O$_2$ (3 bar), 4 h; [b] crotyl alcohol (8.4 mmol), Pd@Au@TiO$_2$ (50 mg), toluene as solvent, O$_2$ (1 atm), 60 °C, 24 h; [c] crotyl alcohol (10 mmol), substrate/catalyst ratio 520:1, O$_2$, 4 h, at 25 °C; [d] 1-octanol (23 mmol), Ru@SQ (50 mg), O$_2$ (10 bar), 100 °C, 36 h; [e] 2-octanol (7.5 mmol), Ru@SQ (50 mg), O$_2$, 115 °C, 60 h; [f] geraniol (0.2 mmol), Au@Zn/Ni-MOF-2 (15 mg), O$_2$ (1 atm), toluene (5 mL), 95 °C, 2 h.

The inverted core–shell, i.e., Au@Pd, was prepared by Balcha et al. using poly(vinylpyrrolidone) (PVP) as a polymer stabilizer and applying both co-reduction and sequential reduction methods. This catalyst was investigated in the aerobic oxidation of crotyl oxidation at room temperature. The conversion and selectivity towards crotonaldehyde after 4 h at 25 °C were 71.4% and 88%, respectively. This composite catalyst was more active than the isolated gold and palladium nanoparticles (conversions of 2.2% and 5.9%, respectively, Entry 4, Table 19) [63].

*4.3. Octanol (C$_8$)*

In 2020, Zhao et al. studied the effect of disodium anthraquinone-2,6-disulfonate (SQ) coating on ruthenium nanoparticles to form the nanocell type Ru@SQ core–shell in the aerobic oxidation of 1-octanol (Scheme 10). The amount of the coating did not affect the substrate conversion (57%), while the selectivity towards the corresponding aldehyde (octanal) increased with the increase in the amount of coating amount reaching 74% (entry 5, Table 19) [47]. The same catalyst was found to be much more efficient for the oxidation of the secondary alcohol (2-octanol) with the conversion and selectivity of 90% and 95% respectively, after 60 h (entry 6, Table 19).

**Scheme 10.** Oxidation of 1-octanol to octanal and 2-octanol to octanone.

### 4.4. Geraniol (C$_{10}$)

Recently, Qin et al. tested Au@Zn/Ni-MOF-2 as a catalyst in the aerobic oxidation of geraniol (Scheme 11) and found that this core–shell is very selective toward geranial (99%), although the conversion after 2 h is quite low (33%, entry 7, Table 19) [54].

**Scheme 11.** Oxidation of geraniol to geranial.

## 5. Oxidation of Alicyclic Alcohols

### 5.1. Cyclohexanol

The catalytic activity of Ru@SQ was tested in the oxidation of various alcohols to carbonyl compounds, and one of the substrates used was cyclohexanol (Scheme 12). High values of the alcohol conversion and selectivity towards cyclohexanone were obtained (entry 1, Table 20) [47].

**Scheme 12.** Oxidation of cyclohexanol to cyclohexanone.

**Table 20.** Catalytic results (yield and selectivity to ketone) for the oxidation of cyclohexanol using different core–shell as catalysts.

| Entry | Catalyst | Yield (%) | Selectivity (%) | Ref. |
|---|---|---|---|---|
| 1 [a] | Ru@SQ | >90 | >90 | [47] |
| 2 [b] | Fe$_3$O$_4$@P4VP@FeCl$_3$ | 22 | 24 | [52] |
| 3 [c] | γ-Fe$_2$O$_3$@Ni$_3$Al-LDH@Au$_{25}$-0.053 | 99 | 99 | [37] |
| 4 [d] | Fe$_3$O$_4$@SiO$_2$@VO(ephedrine)$_2$ | 90 | >99 | [49] |

[a] Reaction conditions: cyclohexanol (10 mmol), Ru@SQ (50 mg), O$_2$ as oxidant, 115 °C, 60 h; [b] cyclohexanol (1 mmol), 2 mol% of Fe$_3$O$_4$@P4VP@FeCl$_3$ catalyst, acetonitrile (5 mL), TEMPO (0.2 mmol), KNO$_2$ (0.2 mmol), 60 °C, 12 h under 1 atm O$_2$; [c] cyclohexanol (1 mmol), γ-Fe$_2$O$_3$@Ni$_3$Al-LDH@Au$_{25}$-0.053 (Au: 0.115%), toluene (5 mL), 80 °C, 4 h under 1 atm O$_2$ (20 mL/min); [d] cyclohexanol (1 mmol), Fe$_3$O$_4$@SiO$_2$@VO(ephedrine)$_2$ (50 mg), polyethylene glycol (1 mL), 80 °C, 12 h and TBHP 70% aq. sol. (4 mmol) as oxidant.

In 2017, Li et al. tested Fe$_3$O$_4$@P4VP@FeCl$_3$ in the aerobic oxidation reaction of cyclohexanol, and this core–shell did not show much catalytic activity with product yield and selectivity to cyclohexanone of 22% and 24%, respectively (entry 2, Table 20) [52].

The best catalytic performance in the cyclohexanol oxidation was demonstrated by the γ-Fe$_2$O$_3$@Ni$_3$Al-LDH@Au$_{25}$-0.053 core–shell composite with molecular oxygen as an oxidant [37] and by the Fe$_3$O$_4$@SiO$_2$@VO(ephedrine)$_2$ core–shell composites with TBHP as an oxidant [49]. The reported conversion and selectivity were 90–99 and 99%, respectively (entries 3 and 4, Table 20).

### 5.2. 3-Methylcycloheptanol

The only core–shell system explored for the oxidation of this secondary alcohol is the Au@Zn/Ni-MOF-2 composite. The aerobic conversion of 3-methylcycloheptanol into the corresponding ketone (Scheme 13) after 2 h at 95 °C is quite low (31%), but the reaction is completely selective [54].

**Scheme 13.** Oxidation of 3-methylheptanol to 3-methylheptanone.

## 6. Reaction Mechanisms

This section will shed some light on the type of mechanism that may be involved in some of the reactions mentioned in the previous sections and may point the reader in the right direction for the development of new core–shell type catalysts.

### 6.1. Benzylic Alcohol

Recently, Su et al. proposed that the oxidation reaction of benzyl alcohol in the liquid phase and in the presence of PILM@Au@Al(OH)$_3$ could occur in two steps, the first involving the adsorption of the alcohol onto the metal center on the surface, forming the metal alkoxide and metal hydride and yielding dehydrobenzyl alcohol by deprotonation, and the second involving the oxidation of the dehydrobenzyl alcohol and metal hydride, regenerating the metal surface [46].

The proposed mechanism for the oxidation of benzyl alcohol with TBHP catalyzed by Fe$_3$O$_4$@SiO$_2$@VO(ephedrine)$_2$ involves the formation of a peroxy intermediate. For the reaction carried out in the presence of 2,2′-azobis(isobutyronitrile), a radical trap, no significant inhibition was found, and therefore, it is unlikely to occur via a radical mechanism [49].

The reaction catalyzed by CoFe$_2$O$_4$@SiO$_2$@[MoO$_2$(salenac-OH)] is also expected to involve the formation of intermediate peroxo species. An initial phase should involve the transfer of a proton from the oxidant *t*-BuOOH to an oxygen atom in MoO$_2$, followed by its coordination to the metal center to form the peroxy intermediate. The peroxy intermediate species will then oxidize benzyl alcohol releasing a *tert*-BuOH molecule, and forming an intermediate that produces benzaldehyde and H$_2$O [55].

In another case, Kong et al. proposed that the mechanism of benzyl alcohol oxidation may involve dehydrogenation of the alcohol at the metal center of the catalyst and the subsequent release of hydrogen upon interaction with the 'activated' oxygen or other species capable of abstracting the adsorbed hydrogen. In this case, the oxygen defect-rich CeO$_2$ layer in the Pd/Fe$_3$O$_4$@mCeO$_2$ core–shell activates oxygen molecules, which releases the adsorbed hydrogen from the metal surface and leads to a faster recovery of the active centers (Figure 6) [51].

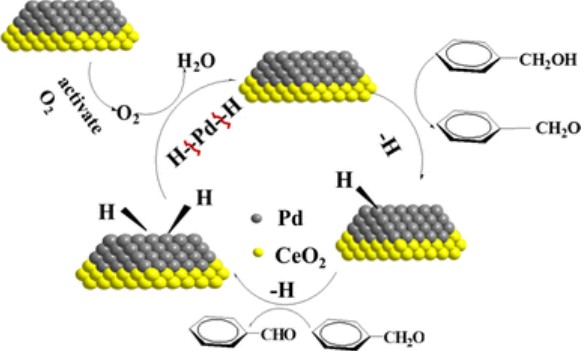

**Figure 6.** Proposed mechanism for benzyl alcohol oxidation. Reproduced with permission from [51].

In order to gain some insight into the type of mechanism involved in the oxidation of benzyl alcohol with Fe$_3$O$_4$@P4VP@FeCl$_3$ as catalyst, Li et al. performed an experiment in the absence of 2,2,6,6-tetramethylpiperidinyl-1-oxy (TEMPO)—a recognized radical initiator—and no benzaldehyde was detected under these conditions, suggesting the involvement of radical species [52]. Furthermore, the use of an O$_2$ atmosphere is also critical for high conversion. The proposed mechanism involves two redox processes promoted by TEMPO and initiated by Fe(III), involving multiple electron and proton transfers (Figure 7). The overall process corresponds to the oxidation of Fe(II) to Fe(III) by NO$_2$ and the oxidation of TEMPOH to TEMPO by Fe(III) [52].

**Figure 7.** Proposed mechanism for the catalytic oxidation of benzyl alcohol. Reproduced from [52] with permission from the Royal Society of Chemistry.

In 2023, Hou et al. also reported the involvement of TEMPO in the mechanism of the aerobic oxidation of benzyl alcohols using $Fe_3O_4@Cu_3(BTC)_2$ core–shell as a catalyst. According to the proposed mechanism, a $Cu^{II}$ superoxide species is formed by the reaction of the $Cu^I$ center with $O_2$. The superoxide $Cu^{II}$ would be oxidized by H-transfer (TEMPOH to TEMPO) to form $Cu^{II}$-OOH. The new $Cu^{II}$-OOH intermediate would then react with water to form the $Cu^{II}$-OH and $H_2O_2$ species (the latter reduced to $H_2O$ and $O_2$ in the presence of Cu). Finally, $Cu^{II}$-OH reacts to form $Cu^{II}$-alkoxide, followed by abstraction of the H atom by TEMPO [56].

*6.2. Vanillic Alcohol*

The ZnO shell with high electron mobility plays an important role in the process, not only inhibiting the leaching of Co species but also promoting a Co(II)/Co(III) redox process. On the other hand, the carbon matrix is responsible for the dispersion of active Co(II/III) species providing more active centers (Figure 8) [59].

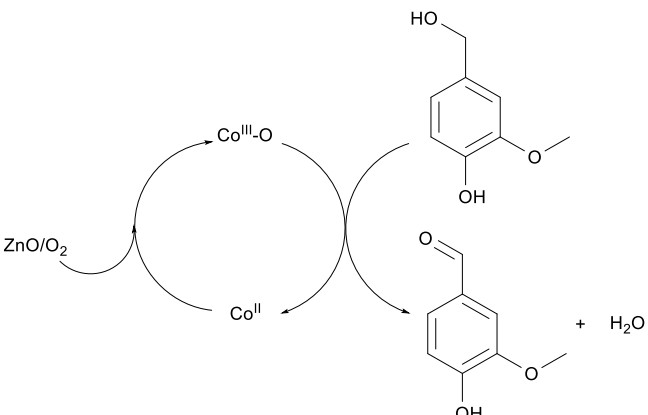

**Figure 8.** Proposed mechanism for the oxidation of vanillic alcohol to vanillin by Co(II/III)@ZnO structured composites.

*6.3. 2,5-Hydroxymethylfurfural*

The mechanism of the reaction should involve the adsorption of the aldehyde group in HMF onto the strong Lewis acid sites ($Ti^{4+}$ species) of the $TiO_2$ coating and its activation by the $OH^-$ ions provided by the base. This results in the formation of a hemiacetal or a gem diol intermediate with subsequent dehydrogenation to a carboxylic acid (Figure 9).

While the $Ti^{3+}/Ti^{4+}$ and $Ce^{4+}/Ce^{3+}$ redox processes occur simultaneously, an $O_2$ molecule is adsorbed on the *N*-doped carbon layer (N/C) of the catalyst, which is rich

in pyridine sites to give the chemisorbed oxygen species such as $O^{2-}$ and $O^-$, the latter becoming part of the $CeO_2$ ($O^{2-}$) network. The gem-diol intermediate then undergoes oxidative dehydrogenation to HMFCA, and a water molecule is formed [60].

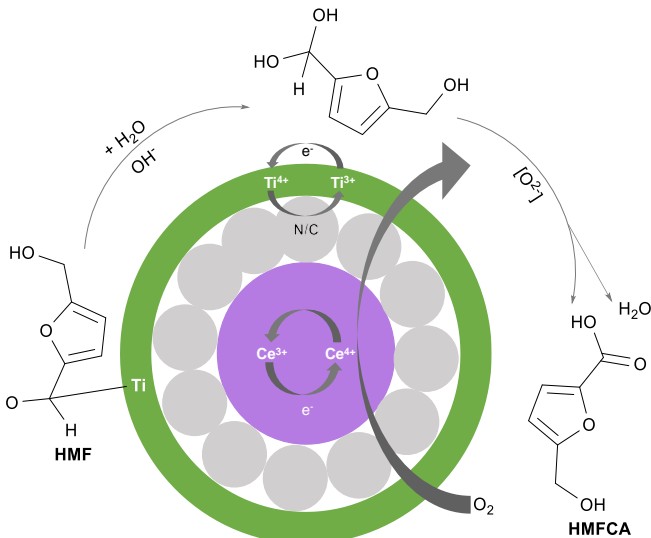

**Figure 9.** Proposed mechanism for the oxidation of 2,5-hydroxymethylfurfural to 5-hydroxymethyl-furan-2-carboxylic acid catalyzed by a D-CeO$_2$@N/C@TiO$_2$ core-shell.

### 6.4. Crotyl Alcohol

One of the possible mechanisms for the selective formation of crotonaldehyde involves a β-H elimination followed by the reaction of $O_2$ with the Pd-H species present on the surface to form water. The other possibility involves a redox process in which the Pd atoms on the surface are oxidized by $O_2$ to form Pd(II) species. These oxidize the alcohol while Pd(II) is reduced back to Pd(0) (Figure 10). Tests were carried out in the presence of the Pd(II) salts and gold nanoparticles, which showed high selectivity towards crotonaldehyde. This observation suggests a cyclic redox mechanism based on the oxidation of crotyl alcohol by the Pd(II) species with the subsequent reduction to the initial Pd(0) state on the surface of Au nanoparticles. These species are oxidized again by the action of $O_2$ and can enter the catalytic cycle again.

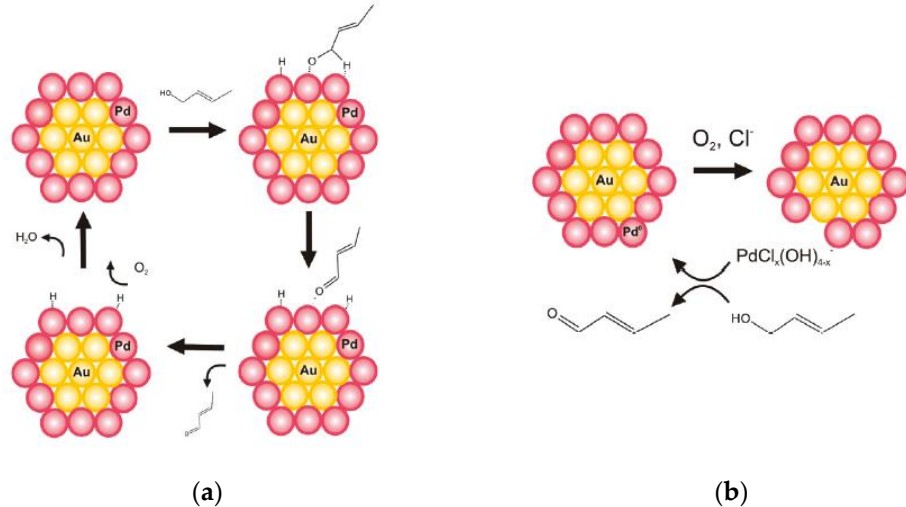

(**a**)  (**b**)

**Figure 10.** Possible mechanisms for the selective aerobic oxidation of crotyl oxidation to crotonalde-hyde by Au@Pd core-shell: (**a**) mechanism involving β-H elimination and reaction of $O_2$ with Pd-H species and (**b**) redox mechanism. Reproduced with permission from [63]. Copyright 2011 American Chemical Society.

## 7. Conclusions

Core–shells are structured nanocomposites that benefit from the combined action of two or more components. Due to the synergistic effect between the different components in the core–shell structures, these composites have improved properties compared to the isolated components. In this paper, we describe recent reports on the use of core–shell composites as catalysts in the conventional oxidation of alcohols. These catalytic systems exhibit particularly high efficiencies in these processes, providing high substrate conversions and reaction selectivities under thermal conditions at temperatures and reaction times suitable for industrial applications. Due to the unique structural features of the core–shell nanocomposites, these species are typically stable under the oxidative conditions, exhibit low leaching, and allow their easy recovery and separation from the reaction mixture and reuse in several catalytic cycles without a significant loss of the catalytic activity. The use of the magnetic materials (e.g., those containing $Fe_2O_3$ or $Fe_3O_4$) further facilitates the catalyst recovery. There is no doubt that the interest in these promising catalytic systems will only increase in the future.

Nevertheless, researchers face several challenges, including the effects of reaction conditions on the morphology and structure of core-shell materials, resorting to advance in situ and real-time observation techniques to understand the structural changes that occur under specific conditions. In addition, the development of synthetic methods that allow the replacement of certain noble metals by others that are cheaper and equally active ones, in order to be economically sustainable and maintain quality and with the aim of scaling-up the synthesis processes themselves, remains a challenge.

**Author Contributions:** Writing—original draft preparation, L.M.M.C.; writing—review and editing, E.C.B.A.A. and M.L.K.; funding acquisition, E.C.B.A.A. All authors have read and agreed to the published version of the manuscript.

**Funding:** This work was partially supported by the Fundação para a Ciência e a Tecnologia (FCT), through projects UIDB/00100/2020 and UIDP/00100/2020 of the Centro de Química Estrutural, and through project LA/P/0056/2020 of the Institute of Molecular Sciences and 2022.02069.PTDC project. Authors are grateful to Instituto Politécnico de Lisboa for the IPL/2022/MMOF4CO2_ISEL project. L.M.M.C. thanks the Fundação para a Ciência e Tecnologia (FCT) for funding his Ph.D. (grant number UI/BD/152790/2022).

**Data Availability Statement:** Data sharing not applicable.

**Conflicts of Interest:** The authors declare no conflict of interest.

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
