# Peer review of "Core–Shell Catalysts for Conventional Oxidation of Alcohols: A Brief Review"

_catalysts, doi:10.3390/catal13071137_

Round 1
Reviewer 1 Report
The paper deals with a review of the study of core-shell catalysts for the conventional oxidation of alcohols, a reaction that plays a fundamental role in organic synthesis, pharmaceutical development, materials science, and various other fields, offering a versatile and indispensable tool for the preparation of valuable compounds and the advancement of scientific knowledge. The development of heterogeneous catalysts for these reactions is important, from the scientific and academic point of view. There are no reviews on this topic published, which will help researchers interested on the theme.
The review is well written and was divided into 6 sessions, including introduction (pages 1-3), oxidation of aromatic homocyclic alcohols (pages 3-18), oxidation of aromatic heterocyclic alcohols (pages 18-20), oxidation of aliphatic alcohols (pages 20-23), oxidation of alicyclic alcohols (pages 23 -24) and conclusions. The article brings a good revision on the topic, with relevant results being discussed. I suggest a few corrections and the insertion of new and relevant papers, already published (selected below). After that, I believe that the review would be suitable to be published in the journal Catalyst.
Page 3, line 84: The authors wrote:
“This review article focuses on the application of the core-shell composites in catalysis, particularly in the conventional oxidation of alcohols into products with added value, such as benzaldehyde, acetophenone, benzophenone, cinnamaldehyde, and vanillin, among others. The oxidation of aromatic homocyclic, aromatic heterocyclic, aliphatic, and alicyclic alcohols catalyzed by the core-shell composite catalysts are discussed in the following four sections”. Most of the review is devoted to the oxidation of aromatic alcohols. Authors must insert a justification or reason for this choice in this paragraph.
New interesting and relevant papers were already published on the theme. Please, insert a short discussion on the selected articles presented below:
-2023 From AuPd Nanoparticle Alloys towards Core-Shell Motifs with Enhanced Alcohol Oxidation Activity https://doi.org/10.1002/cctc.202300180
2023 Rational Design of Highly Efficient PdIn–In2O3 Interfaces by a Capture-Alloying Strategy for Benzyl Alcohol Partial Oxidation https://doi.org/10.1021/acsami.3c00810
-2023 Synthesis of a Magnetic Core-shell Fe3O4@Cu-3(BTC)(2) Catalyst and Its Application in Aerobic Olefin Epoxidation, Chemical Research in Chinese Universities (2023) (the main topic is epoxidation, but results on alcohol oxidation are also discussed https://doi.org/10.1007/s40242-023-3106-1
-2023 Synthesis of a Novel Spherical-Shell-Structure Polymerized Ionic Liquid Microsphere PILM/Au/Al(OH)3 Catalyst for Benzyl Alcohol Oxidation https://doi.org/10.1021/acsami.2c20967
-2022 Construction of a nitrogen doped graphene-wrapped Fe3O4@polydopamine/Pd core–shell nanooctahedral for enhanced reduction of nitroarenes and oxidation of alcohols
https://doi.org/10.1016/j.solidstatesciences.2022.107026
-2022 Solvent-free selective oxidation of alcohols with tert-butyl hydroperoxide catalyzed by dioxo-molybdenum (VI) unsymmetrical Schiff base complex immobilized on CoFe2O4@SiO2 nanoparticles https://doi.org/10.1007/s11164-022-04835-1
Author Response
Reviewer 1
We thank this Reviewer for his/her valuable comments and suggestions which helped us to improve the manuscript.
Reviewer 1 wrote: “The paper deals with a review of the study of core-shell catalysts for the conventional oxidation of alcohols, a reaction that plays a fundamental role in organic synthesis, pharmaceutical development, materials science, and various other fields, offering a versatile and indispensable tool for the preparation of valuable compounds and the advancement of scientific knowledge. The development of heterogeneous catalysts for these reactions is important, from the scientific and academic point of view. There are no reviews on this topic published, which will help researchers interested on the theme.
The review is well written and was divided into 6 sessions, including introduction (pages 1-3), oxidation of aromatic homocyclic alcohols (pages 3-18), oxidation of aromatic heterocyclic alcohols (pages 18-20), oxidation of aliphatic alcohols (pages 20-23), oxidation of alicyclic alcohols (pages 23 -24) and conclusions. The article brings a good revision on the topic, with relevant results being discussed. I suggest a few corrections and the insertion of new and relevant papers, already published (selected below). After that, I believe that the review would be suitable to be published in the journal Catalyst.
Page 3, line 84: The authors wrote:
“This review article focuses on the application of the core-shell composites in catalysis, particularly in the conventional oxidation of alcohols into products with added value, such as benzaldehyde, acetophenone, benzophenone, cinnamaldehyde, and vanillin, among others. The oxidation of aromatic homocyclic, aromatic heterocyclic, aliphatic, and alicyclic alcohols catalyzed by the core-shell composite catalysts are discussed in the following four sections”. Most of the review is devoted to the oxidation of aromatic alcohols. Authors must insert a justification or reason for this choice in this paragraph.”
Our response: The structure of this review reflects the much lower number of publications about the oxidation of aliphatic alcohols compared to the aromatic ones. This, in turn, is due to the relatively higher activity and selectivity of the latter compared to the former. We inserted this explanation on page 3, lines 88-94.
Reviewer 1 wrote: “New interesting and relevant papers were already published on the theme. Please, insert a short discussion on the selected articles presented below:
-2023 From AuPd Nanoparticle Alloys towards Core-Shell Motifs with Enhanced Alcohol Oxidation Activity https://doi.org/10.1002/cctc.202300180
-2023 Rational Design of Highly Efficient PdIn–In2O3 Interfaces by a Capture-Alloying Strategy for Benzyl Alcohol Partial Oxidation https://doi.org/10.1021/acsami.3c00810
-2023 Synthesis of a Magnetic Core-shell Fe3O4@Cu-3(BTC)(2) Catalyst and Its Application in Aerobic Olefin Epoxidation, Chemical Research in Chinese Universities (2023) (the main topic is epoxidation, but results on alcohol oxidation are also discussed https://doi.org/10.1007/s40242-023-3106-1
-2023 Synthesis of a Novel Spherical-Shell-Structure Polymerized Ionic Liquid Microsphere PILM/Au/Al(OH)3 Catalyst for Benzyl Alcohol Oxidation https://doi.org/10.1021/acsami.2c20967
-2022 Construction of a nitrogen doped graphene-wrapped Fe3O4@polydopamine/Pd core–shell nanooctahedral for enhanced reduction of nitroarenes and oxidation of alcohols. https://doi.org/10.1016/j.solidstatesciences.2022.107026
-2022 Solvent-free selective oxidation of alcohols with tert-butyl hydroperoxide catalyzed by dioxo-molybdenum (VI) unsymmetrical Schiff base complex immobilized on CoFe2O4@SiO2 nanoparticles https://doi.org/10.1007/s11164-022-04835-1”
Our response: As suggested by Reviewer 1, most of the articles were added to the manuscript as Refs. 45, 46, 55, 56, 57, and discussed in the corresponding sections:
Ref. 45: page 5 and lines 195-200.
Ref. 46: page 6 and lines 222-231.
Ref. 55: pages 11/12, lines 415-426 and Table 10.
Ref. 56: page 12, lines 432-437 and Table 11.
Ref. 57: pages 12/13, lines 443-455 and Table 12.

Reviewer 2 Report
The manuscript entitled “Core-Shell Catalysts for Conventional Oxidation of Alcohols: A Brief Review” reviewed the state-of-the-art research on the application of core-shell structured materials as catalysts in the oxidation of alcohols to value-added products. Further, this work also highlights some of the unique advantages in the catalysis of core-shell nanomaterials. However, an attractive review should not only be a summary of the literatures but also has analysis for the literature with clear logic. Hence, a major revision is suggested. Some comments are as below.
1. The current structure of article was described in an order based on the type of alcohols. However, the contents of 2.1 are massive and complex, a deeper understanding for the reaction should be exhibited, such as the features of different types of catalyst includes inorganic@inorganic, inorganic@organic, organic@inorganic and organic@organic combinations.
2. Substrate scope of different alcohols shown in Table 2-9 are repetitive and space-wasting, it might be concisely summarized by words.
3. The style of structural formula in schemes and figures needs to be corrected into a more concise and space-efficient style like ACS style.
4. More discussion about the outlook and future challenges of core-shell catalysts is needed in the Conclusions part, which might enlighten the study of readers in this area.
Moderate editing of English language required
Author Response
Reviewer 2
We thank this Reviewer for his/her valuable comments and suggestions which helped us to improve the manuscript.
Reviewer 2 wrote: “The manuscript entitled “Core-Shell Catalysts for Conventional Oxidation of Alcohols: A Brief Review” reviewed the state-of-the-art research on the application of core-shell structured materials as catalysts in the oxidation of alcohols to value-added products. Further, this work also highlights some of the unique advantages in the catalysis of core-shell nanomaterials. However, an attractive review should not only be a summary of the literatures but also has analysis for the literature with clear logic. Hence, a major revision is suggested. Some comments are as below.
- The current structure of article was described in an order based on the type of alcohols. However, the contents of 2.1 are massive and complex, a deeper understanding for the reaction should be exhibited, such as the features of different types of catalyst includes inorganic@inorganic, inorganic@organic, organic@inorganic and organic@organic combinations.”
Our response: We appreciate the reviewer's comment, however, as can be seen from Table 1, most of the catalysts belong to the inorganic@inorganic combination category so splitting 2.1 considering this classification would not significantly change its content. In lines 254-260, a text has been added regarding the activity of monometallic shells vs. polymetallic core-shells.
Reviewer 2 wrote: “2. Substrate scope of different alcohols shown in Table 2-9 are repetitive and space-wasting, it might be concisely summarized by words.”
Our response: In our opinion, presenting the data in the form of tables makes it easier to correlate them than just describing them in text form. Given that Catalysts is an electronic journal, the issue of wasted space is not so crucial, and we prefer to keep the Tables.
Reviewer 2 wrote: “3. The style of structural formula in schemes and figures needs to be corrected into a more concise and space-efficient style like ACS style.”
Our response: Structures were improved as recommended by the reviewer.
Reviewer 2 wrote: “4. More discussion about the outlook and future challenges of core-shell catalysts is needed in the Conclusions part, which might enlighten the study of readers in this area.”
Our response: As suggested, a comment was added to the conclusions.

Reviewer 3 Report
The overall logic of this article is clear and very well summarised. The advantages and characteristics of nuclear shell materials as catalysts are described in detail.
There are some problems, which should be solved before it is considered for publication.
First of all, Some discussion of the mechanism needs to be added to your article so that the reader can gain a deeper understanding of the article. I suggest that the section "Discussion of the mechanism" be added before "CONCLUTION" in the article.
Secondly, the clarity of some of the images in your article could be spruced up and improved, and the formatting of some tables should be adjusted appropriately. the paper's grammar and academic expression are rough. It is recommended to further enhance the clarity of the picture and polish the language, otherwise, it will be difficult to attract readers.
You should read more literature to expand on the “INTRODUCTION” of this article so that it is of interest to the reader. It is recommended to include "Green Energy & Environment, 2022. doi:10.1016/j.gee.2022. 01.005" in your article.
Given a satisfactory response from the authors, I would propose that the paper be accepted.
You will need to revise your English properly.
Author Response
Reviewer 3
We thank this Reviewer for his/her valuable comments and suggestions which helped us to improve the manuscript.
Reviewer 3 wrote: “The overall logic of this article is clear and very well summarised. The advantages and characteristics of nuclear shell materials as catalysts are described in detail.
There are some problems, which should be solved before it is considered for publication.
First of all, Some discussion of the mechanism needs to be added to your article so that the reader can gain a deeper understanding of the article. I suggest that the section "Discussion of the mechanism" be added before "CONCLUTION" in the article.”
Our response: In accordance with the Reviewer´s comment, we inserted the section “6. Reaction Mechanisms” where the mechanistic details available in the literature for the reactions under study are discussed.
Reviewer 3 wrote: “Secondly, the clarity of some of the images in your article could be spruced up and improved, and the formatting of some tables should be adjusted appropriately. the paper's grammar and academic expression are rough. It is recommended to further enhance the clarity of the picture and polish the language, otherwise, it will be difficult to attract readers.”
Our response: We performed extensive linguistic revision of the text and tried to improve the quality of the images and tables.
Reviewer 3 wrote: “You should read more literature to expand on the “INTRODUCTION” of this article so that it is of interest to the reader. It is recommended to include "Green Energy & Environment, 2022. doi:10.1016/j.gee.2022.01.005" in your article.”
Our response: We inserted several recent works suggested by Reviewer 1 such as Refs. 45, 46, 55–57. Besides that, references 29-31 were also added. However, regarding the reference suggested by the reviewer, we do not see how to insert it in the text since it does not involve the application of catalysts with a core-shell structure and does not refer to the reaction reported in this review, the oxidation of alcohols.

Round 2
Reviewer 2 Report
The comments have been adressed well, and thus I recommend that it can be accepted.